# Scaling Sparse Feature Circuits For Studying In-Context Learning

**Dmitrii Kharlapenko** [* 1]  **Stepan Shabalin** [* 2]  **Arthur Conmy** [3]  **Neel Nanda** [3]

## Abstract

Sparse autoencoders (SAEs) are a popular tool for interpreting large language model activations, but their utility in addressing open questions in interpretability remains unclear. In this work, we demonstrate their effectiveness by using SAEs to deepen our understanding of the mechanism behind in-context learning (ICL). We identify abstract SAE features that (i) encode the model's knowledge of which task to execute and (ii) whose latent vectors causally induce the task zero-shot. This aligns with prior work showing that ICL is mediated by task vectors. We further demonstrate that these task vectors are well approximated by a sparse sum of SAE latents, including these task-execution features. To explore the ICL mechanism, we scale the sparse feature circuits methodology of Marks et al. (2024) to the Gemma 1 2B model for the more complex task of ICL. Through circuit finding, we discover task-detecting features with corresponding SAE latents that activate earlier in the prompt, that detect when tasks have been performed. They are causally linked with task-execution features through the attention and MLP sublayers.

## 1. Introduction

Sparse autoencoders (SAEs; Ng (2011); Bricken et al. (2023); Cunningham et al. (2023)) are a promising method for interpreting large language model (LLM) activations. However, the full potential of SAEs in interpretability research remains to be explored, since most recent SAE research either i) interprets a single SAE's features rather than the model's computation as a whole Bricken et al. (2023), or ii) performs high-level interventions in the model, but does not interpret the effect on the downstream computation

caused by the interventions Templeton et al. (2024). In this work, we address these limitations by interpreting in-context learning (ICL), a widely studied phenomenon in LLMs. In summary, we show that SAEs a) enable the discovery of novel circuit components (**task-detection features**; Section 4.2) and b) refine existing interpretations of ICL, by e.g. decomposing task vectors Todd et al. (2024); Hendel et al. (2023) into **task-execution features** (Section 3).

In-context learning (ICL; Brown et al. (2020)) is a fundamental capability of large language models that allows them to adapt to new tasks without fine-tuning. ICL is a significantly more complex and important task than other behaviors commonly studied in circuit analysis (such as IOI in Wang et al. (2022) and Kissane et al. (2024), or subject-verb agreement and Bias-in-Bios in Marks et al. (2024)). Recent work by (Todd et al., 2024) and Hendel et al. (2023) has introduced the concept of **task vectors** to study ICL in a simple setting, which we follow throughout this paper.[1] In short, task vectors are internal representations of simple operations performed by language models during the processing of few-shot prompts, e.g. as "hot → cold, big → small, fast → slow". We call these simple operations **tasks**. Task vectors can be extracted and added into different LLM forward passes to induce 0-shot task behavior, e.g. making LLMs predict that "slow" follows "fast →" without explicit context. Section 2.3 provides a full introduction.

Unfortunately, it is not possible to decompose task vectors by naively applying sparse autoencoders to reconstruct them. This is because task vectors are out-of-distribution for SAEs and empirically, their SAE decompositions tend to be cluttered with irrelevant features. To address this limitation, we developed the TASK VECTOR CLEANING (**TVC**) algorithm (Section 3.1), which enabled us to extract **task-execution features**: SAE latents that preserve most of the task vector's effect on task loss while being more interpretable. These task-execution features demonstrated two key properties: they could partially substitute for complete task vectors in steering experiments, and they often exhibited clear task-related patterns in their maximum-activating tokens. Through extensive testing across diverse tasks, we

---

[*]Equal contribution  [1]ETH Zurich, Switzerland [2]Georgia Institute of Technology, US [3]Joint senior authors. Correspondence to: Dmitrii Kharlapenko <dkharlapenko@ethz.ch>, Stepan Shabalin <sshabalin3@gatech.edu>.

*Proceedings of the 42$^{nd}$ International Conference on Machine Learning*, Vancouver, Canada. PMLR 267, 2025. Copyright 2025 by the author(s).

---

[1]Task vectors Hendel et al. (2023) are also called "function vectors" Todd et al. (2024), but we use "task vectors" throughout this paper for consistency.

validated that these features play a causal role in the model's ICL capabilities (Section 3.2).

To understand how these features operate within the broader ICL mechanism, we extended the Sparse Feature Circuits (SFC) discovery algorithm introduced by Marks et al. (2024) (Section 4). Applying this algorithm to the Gemma 1 2B model Team et al. (2024) revealed a network of SAE latents crucial to ICL processing. This analysis uncovered a complementary set of features we call **task-detection features**, which identify the specific task being performed based on earlier prompt information. Importantly, our experiments demonstrated that disabling task-detection feature directions also disabled task-execution directions, revealing their fundamental interdependence in the ICL circuit (Section 4.2).

Our findings not only advance our understanding of ICL mechanisms but also demonstrate the potential of SAEs as a powerful tool for interpretability research on larger language models. By unifying the task vectors view with SAEs and uncovering two of the most important causally implicated feature families behind ICL, we pave the way for future work to control and monitor ICL further, to improve either the safety or capabilities of models.

Our main contributions are as follows:

1. We demonstrate that SAEs can be effectively used to analyze a complex ICL mechanism in Gemma 1 2B, which is **one of the largest models** studied at this depth in comparable, end-to-end circuits-style mechanistic interpretability research Wang et al. (2022); Marks et al. (2024) (Section 4.1.3).

2. We identify **two core components of the ICL circuit** in Gemma 1 2B: task-detection features that identify required tasks from the prompt, and task-execution features that implement those tasks during generation (Sections 3 and 4.2).

3. We uncover how these components interact: attention heads and MLPs process information from task-detection features to activate the appropriate task-execution features, **revealing how the model integrates information across the prompt to perform ICL tasks** (Section 4.2).

4. We develop the Task Vector Cleaning (TVC) algorithm, a **novel transformer-specific sparse linear decomposition method** that enables precise analysis of ICL mechanisms by decomposing task vectors into their most relevant features (Section 3.1).

## 2. Background

### 2.1. Sparse Autoencoders (SAEs)

Sparse autoencoders (SAEs) are neural networks designed to learn efficient representations of data by enforcing sparsity

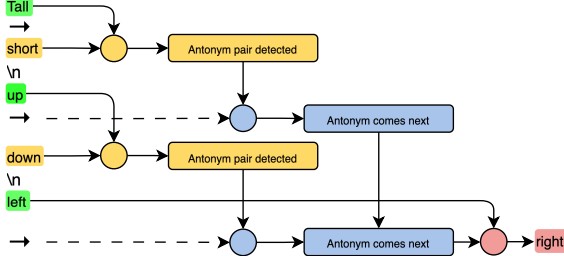

Figure 1. A diagram of the in-context learning circuit, showing task detection features (yellow) causing task execution features (blue) which cause the model to output the antonym (left → right). A more concrete circuit, along with texts these features activate on, can be seen in Figure 12.

in the hidden layer activations Elad (2010). In the context of language model interpretability, SAEs are used to decompose the high-dimensional activations of language models into more interpretable features (Cunningham et al., 2023; Bricken et al., 2023). The basic idea behind SAEs is to train a neural network to reconstruct its input while constraining the hidden layer to have sparse activations. This process typically involves an encoder that maps the input to a sparse hidden representation, a decoder that reconstructs the input from this sparse representation, and loss task that balances reconstruction accuracy with sparsity.[2] The encoding step is as follows, with $\mathbf{f}$ denoting the pre-activation features and $\mathbf{W}_{\text{enc}}$ and $\mathbf{b}_{\text{enc}}$ the encoder weights and biases respectively:

$$\mathbf{f}(\mathbf{x}) = \sigma(\mathbf{W}_{\text{enc}}\mathbf{x} + \mathbf{b}_{\text{enc}}) \quad (1)$$

For JumpReLU SAEs Rajamanoharan et al. (2024b), the activation function and decoder are (with $H$ being the Heaviside step function, $\theta$ the threshold parameter and $\mathbf{W}_{\text{dec}}$/$\mathbf{b}_{\text{dec}}$ the decoder affine parameters):

$$\hat{\mathbf{x}}(\mathbf{f}) = \mathbf{W}_{\text{dec}}(\mathbf{f} \odot H(\mathbf{f} - \theta)) + \mathbf{b}_{\text{dec}} \quad (2)$$

In our work, we train SAEs on residual stream activations and attention outputs, and also train transcoders[3] on MLP layers, all of which use the improved Gated SAE architecture Rajamanoharan et al. (2024a).

---

[2]Typically, the $L_1$ penalty on activations is used Bricken et al. (2023) with some modifications Rajamanoharan et al. (2024a); Conerly et al. (2024), although there are alternatives: Rajamanoharan et al., 2024b; Farrell, 2024; Riggs & Brinkman, 2024.

[3]Transcoders (Dunefsky et al., 2024) are a modification of SAEs that take MLP input and convert it into MLP output instead of trying to reconstruct the residual stream.

## 2.2. Sparse Feature Circuits

Sparse Feature Circuits (SFCs) are a methodology introduced by Marks et al. (2024) to identify and analyze causal subgraphs of sparse autoencoder features that explain specific model behaviors. This approach combines the interpretability benefits of SAEs with causal analysis to uncover the mechanisms underlying language model behavior. The SFC methodology involves several key steps:

1. Decomposing model activations into sparse features using SAEs

2. Calculating the Indirect Effect (IE, (Pearl, 2001) of each feature on the target behavior

3. Identifying a set of causally relevant features based on IE thresholds

4. Constructing a circuit by analyzing the connections between these features

The IE of a model component is measured by intervening on that component and observing the change in the model's output. For a component $a$ and a metric $m$, the IE is defined using do-calculus Pearl (2009) as in (Marks et al., 2024) as:

$$\text{IE}(m; a) = m(x|\text{do}(a = a')) - m(x) \qquad (3)$$

Where $m(x|\text{do}(a = a'))$ represents the value of the metric when we intervene to set the value of component $a$ to $a'$, and $m(x)$ is the original value of the metric. In practice, attribution patching Syed et al. (2023) is used to approximate IE, allowing for efficient computation across many components simultaneously.

SFC is described in detail in Marks et al. (2024). We describe our modifications in Section 4 and Appendix E.

## 2.3. Task Vectors

Continuing from the high-level description in Section 1, task vectors were independently discovered by Hendel et al. (2023) and Todd et al. (2024). The key idea behind task vectors is that they capture the essence of a task demonstrated in a few-shot prompt, allowing the model to apply this learned task to new inputs without explicit fine-tuning. Task vectors have several important properties:

1. They can be extracted from the model's hidden states given ICL prompts as inputs.
2. When added to the model's activations in a zero-shot setting, they can induce task performance without explicit context.
3. They appear to encode abstract task information, independent of specific input-output examples.

To illustrate the concept, consider the following simple prompt for an antonym task in the Example 1, where boxes represent distinct tokens:

BOS  Follow  the  pattern  :  \n
hot  →  cold  \n
big  →  small  \n
fast  →  slow

*Example 1.* All token types in an example input: prompt , input , arrow , output , newline ( target tokens for calculating the loss on included).

In this case, the task vector would encode the abstract notion of "finding the antonym" rather than specific word pairs. Task vectors are typically collected by averaging the residual stream of "→" tokens at a specific layer across multiple ICL prompts for a given task. This averaged representation can then be used to study the model's internal task representations and to manipulate its behavior in zero-shot settings. We perform our analysis on the datasets derived from Todd et al. (2024). Details can be found in Appendix A.

As noted in the task vector paper, the impact of task vectors on loss varies across model layers (Figure 2). Middle layers show the strongest effects, while earlier and later layers demonstrate reduced effectiveness. For our experiments, we selected a single optimal layer - termed the *target layer* - from these high-performing middle layers. In the Gemma 1 2B experiments, layer 12 served as this target layer.

## 3. Discovering Task-Execution Features

### 3.1. Decomposing task vectors

To gain a deeper understanding of task vectors, we attempted to decompose them using sparse autoencoders (SAEs). However, several of our initial naive approaches faced significant challenges. Firstly, direct SAE reconstruction, i.e. passing the task vector as input to the SAE, produced noisy results with more than 10 non-zero SAE features on average on target layers (in Gemma 1 2B), most of which were irrelevant to the task. Moreover, this reconstruction noticeably reduced the vector's effect on task loss. These issues arose partly because task vectors are out-of-distribution inputs for SAEs, as they aggregate information from different residual streams rather than representing a single one.

We then explored inference-time optimization (ITO; Smith (2024)) as an alternative. However, this method also failed to reconstruct task vectors using a low number of SAE features while maintaining high effect the loss.

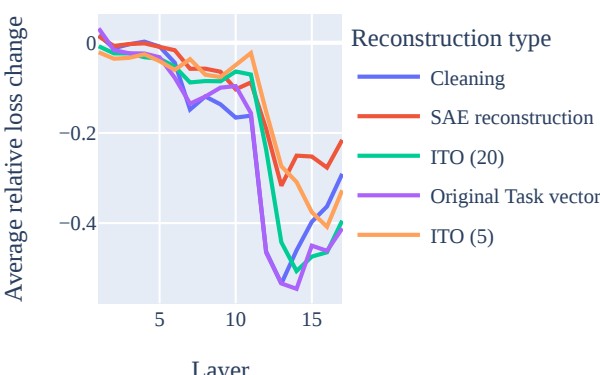

Figure 2. The effect on the Gemma 1 2B's task losses by steering with different kinds of reconstructed task vectors, at each layer. We see that cleaning performs similarly to the original task vector until layer 14. Average relative loss change measured as a post-steering relative loss change compared to 0-shot, averaged across all tasks.

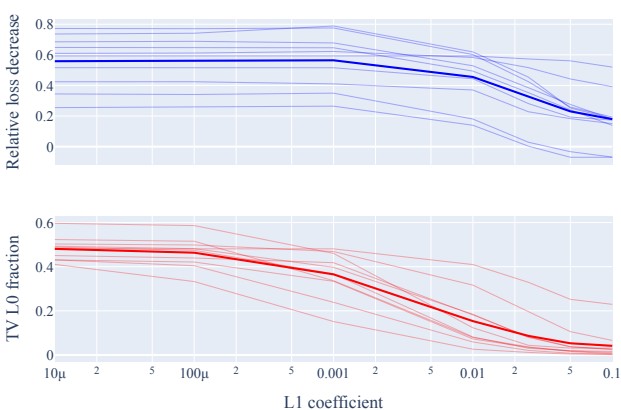

Figure 3. Evaluation of task vectors after applying our TVC algorithm across different $L_1$ coefficient $\lambda$ values. Top: relative decrease in loss after steering (higher → better); bottom: fraction of retained active features (lower → better). Transparent lines represent different model and SAE combinations; solid lines show means across all of them; the results are averaged across tasks; (x-axis: $L_1$ coefficient $\lambda$). Further details in Appendix D.1.

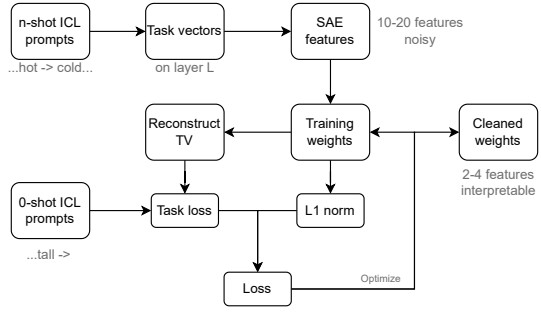

Figure 4. Overview of the task vector cleaning algorithm; TV stands for task vector.

Given these observations, we developed a novel method called **task vector cleaning** (Figure 4). It produces optimized SAE decomposition weights $\theta \in \mathbb{R}^{d_{SAE}}$ for a task vector $v_{tv}$. At a high level, the method:

1. Initializes $\theta$ with weights from SAE decomposition of $v_{tv}$

2. Reconstructs a new task vector $v_\theta$ from $\theta$; steers the model with $v_\theta$ on a batch of zero-shot prompts and computes negative log-likelihood loss $\mathcal{L}_{NLL}(\theta)$ on them

3. Optimizes $\theta$ to minimize $\mathcal{L} = \mathcal{L}_{NLL}(\theta) + \lambda\|\theta\|_1$, where $\lambda$ is the sparsity coefficient

This approach allows us to maintain or even improve the effect on the loss of task vectors while reducing the amount of active SAE features to by 70% (Figure 3) for Gemma 1 2B and other models. The algorithm overview can be found in Figure 4. More details are provided in Appendix D.

We evaluated TVC against four baseline approaches: (1) original task vectors, (2) naive SAE reconstruction, (3) ITO with target $L_0$ norm of 5, and (4) ITO with target $L_0$ norm of 20. The evaluation methodology involved steering zero-shot prompts using reconstructed task vectors and measuring the relative improvement in log-likelihood loss, averaged across all tasks. The steering is done in the same manner as in the TVC algorithm, further details can be found in Appendix D. Figure Figure 2 presents the layer-wise comparison results. To validate robustness, we conducted extensive parameter sweeps of the $L_1$ regularization coefficient $\lambda$ across multiple model scales, architectures and SAE configurations, including various widths and target sparsities for both Gemma-2 2B and 9B models, with the aggregated results shown in Figure 3. As detailed in Appendix D.1, these experiments demonstrated that TVC consistently achieves a 50-80% reduction in active SAE features while maintaining the task vectors' effect on loss. The results further indicated enhanced performance when using SAEs with higher target $L_0$ values.

This decomposition approach revealed interpretable features that clearly corresponded to their intended tasks. Most notably, we identified a class of features we termed *"task-execution features"* (or "executor features"), characterized by three properties:

1. They activate upon encountering task-relevant examples in natural text.
2. Their activation peaks on the token immediately preceding task completion.
3. They have a high causal task-specific effect, which we measure later.

To illustrate, consider an antonym task feature processing the phrase " hot and cold ." The feature activates on " and ," indicating the model's anticipation of an upcoming antonym. This suggests the model recognizes antonym relationships before observing the complete pair. Figure 5 provides additional examples of such features. Further examples in Appendix I demonstrate these features' maximum activations on SAE training data, consistently showing task-specific activation patterns.

import and domestic, UI    ecial joie de vivre (joy of life

etween fresh and traditional,    ) Banderitas (little paper bann

both local and remote event    ıdad dorada" (the golden city

th tropical and temperate wa    jihad, or struggle, in

(a) Antonyms executor feature 11618.  (b) Translation to English executor feature 5579.

*Figure 5.* A subset of max activating examples for executor features from Appendix I.

To analyze the activation patterns of executor features, we split all ICL prompt tokens into several types (highlighted in Example 1 and discussed later in Section 4.1.1). For each executor feature, we calculate its token type *activation masses*: the sum of all its activations on tokens of a particular type across a batch of ICL prompts. Table 1 shows the percentages of total mass split among different token types for executor features. We can see that executors activate largely on arrow tokens.

### 3.2. Steering Experiments

To validate the causal relevance of our decomposed task features, we conducted a series of steering experiments, observing the features' impact on task loss across different contexts and model layers.

The experiments were performed on the dataset of diverse tasks taken from Todd et al. (2024). We first extracted relevant task features using our cleaning algorithm. Then steered the zero-shot prompt using them and calculated relative loss improvement, normalizing and clipping it after that. In this section we present main results from the Gemma 1 2B model. Further details and additional experiments that include other models can be found in Appendix F.

Figure 6 shows a heatmap of steering results for each pair of tasks and task-relevant features. Higher values indicate greater improvement in the loss after steering. It can be seen that most features that have a high effect on some task generally do not significantly affect unrelated tasks. Another notable detail is that features from related tasks (like the translation group) at least partially affect all tasks within the group.

We have manually examined the features with the highest effect and found that their maximum activating dataset examples tend to align with their hypothesized role in the ICL circuit. Interestingly, we observed that tasks requiring translation from English all share a generic English-to-foreign language task execution feature. This shared feature suggests a common mechanism for translation tasks, with language-specific information encoded separately. Max activating examples of the most interpretable features are present in Appendix I.

We also observed that with high $L_0$ constraints TVC may drop this common translation feature in favor of a combination of other multi-language features, that have weaker effect individually. This did not happen with English-to-Spanish, so we attribute this to biases in SAE and Gemma training data, since Spanish is usually much better represented than French or Italian. This is supported by to-French and to-Italian translation tasks having much higher loss on their ICL prompts than to-Spanish.

| Token Type | Mass (%) |
|---|---|
| arrow | 89.80 |
| output | 6.46 |
| input | 3.2 |
| newline | 0.54 |
| prompt | 0.00 |

*Table 1.* Activation masses for **executor** features across different token types, averaged across all tasks. We can notice they activate largely on arrow tokens.

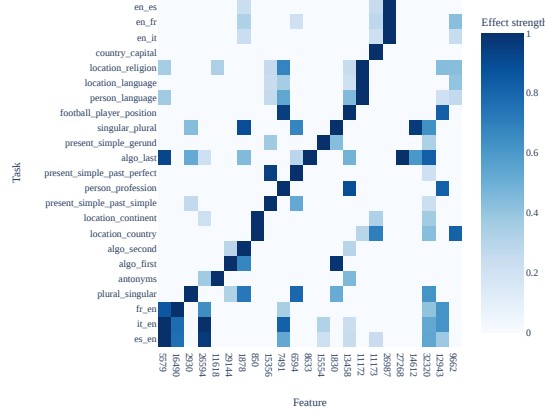

*Figure 6.* Heatmap showing the effect of steering with individual task-execution features for each task. Most features boost exactly one task, with a few exceptions for similar tasks like translating to English. Full and unfiltered versions of the heatmap are available in Appendix F.

# 4. Applying SFC to ICL

After identifying task-execution features through our task vector analysis, we sought to expand our understanding of the in-context learning (ICL) circuit. To this end, we apply the Sparse Feature Circuits (SFC) methodology Marks et al. (2024) to the Gemma 1 2B model. However, due to the increased complexity of ICL tasks and the larger model size, the original SFC approach did not work out of the box. We had to implement several key modifications to address the challenges we encountered.

## 4.1. Our Modifications

### 4.1.1. TOKEN POSITION CATEGORIZATION AND FEATURE AGGREGATION

We modified the SFC approach to better handle the structured nature of ICL prompts. Instead of treating each SAE feature as a separate node, we categorized token positions into the following groups:

- Prompt : The initial instruction tokens (e.g., "Follow the pattern:")

- Input : The last token before each arrow in an example pair

- Arrow : The arrow token itself ("→")

- Output : The last token before each newline in an example pair

- Newline : The newline token

- Extra: Any tokens not covered by the above categories (e.g., in multi-token inputs or outputs)

Each pair of an SAE feature and a token type was assigned its own graph node. The effects of the feature were aggregated across all tokens of the corresponding type. This categorization allowed us to evaluate how features affect all tokens within the same category, separating features based on their role in the ICL circuit. It also enabled us to selectively disable parts of the circuit for one task while testing another, verifying the task specificity of the identified circuits.

### 4.1.2. LOSS FUNCTION MODIFICATION

An ICL prompt can be viewed as an $(x, y)$ pair, where $x$ represents the entire prompt except for the last pair's output, and $y$ represents this output. The original SFC paper suggested using the log probabilities of $y$ conditioned on $x$ for such datasets. However, this approach often resulted in task-relevant features having high negative IEs on other example pairs in the prompt. This was likely due to the

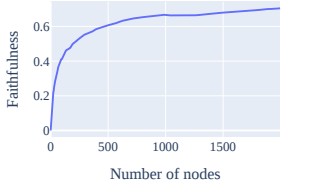
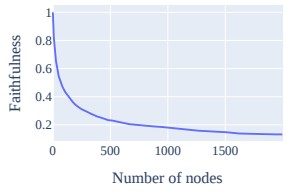

(a) Faithfulness of $C$        (b) Faithfulness of $M \setminus C$

*Figure 7.* Faithfulness for task circuits $C$ (a) and their complements $M \setminus C$ (b). The ideal score for $C$ is 1, and 0 for $M \setminus C$.

circuit's effect on those pairs being lost to either diminishing gradients in backpropagation or because copying circuits were much more relevant to predicting the last pair. By considering all pairs except the first one, we amplified the effect of the task-solving circuit relative to the numerous cloning circuits that activate due to the repetitive nature of ICL prompts.

### 4.1.3. SFC EVALUATION

To evaluate our SFC modifications, we measured faithfulness through ablation studies on our ICL task dataset. Following Marks et al. (2024), we define faithfulness as:

$$\text{Faithfulness}(C) = \frac{m(C) - m(\varnothing)}{m(M) - m(\varnothing)} \quad (4)$$

where $m(C)$ is the metric with circuit $C$, $m(\varnothing)$ represents the empty circuit baseline, and $m(M)$ is the complete model's performance. While Marks et al. (2024) uses mean ablation, we opt for zero ablation since it better aligns with the sparse nature of SAE features.

This metric quantifies how much of the model's original capabilities are preserved when isolating specific circuit components.

To measure faithfulness, we first approximated node Indirect Effects (IEs) using a batch of few-shot examples for a given task. We then evaluated ablation effects on a held-out batch from the same task to ensure our findings generalize beyond the examples used for IE estimation. Our ablation studies targeted both the discovered circuit $C$ and its complement $M \setminus C$, removing nodes according to their IE thresholds. Figure 7 shows that circuits of 500 nodes were enough to achieve an average faithfulness of 0.6 across tasks, matching the performance seen in complex cases from the SFC paper. While the original SFC paper required only tens of top features disabled (Faithfulness of $M \setminus C$) to impact task performance, our approach needed several hundred features to achieve similar degradation. Should we focus ablation just on layers 11-12, we need much less active nodes for faithfulness 0.6 – around 10-60 on average.

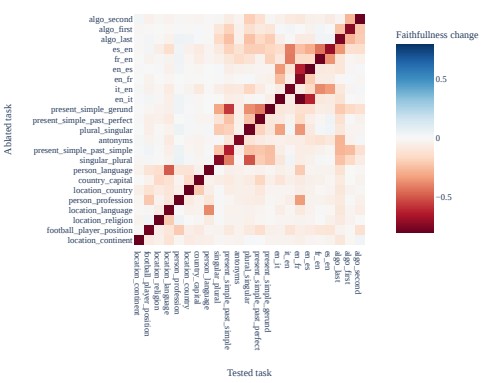

*Figure 8.* We study how useful the most important nodes on task A are for performance on task B. Specifically, we ablate the most important features for task A (the ablated task on the $y$-axis) so that faithfulness reduces by 0.7, and measure how much faithful reduces on another task B (the tested task on the $x$-axis).

We need to note that we constrained our circuit search to intermediate layers 10-17 of the 18 total layers. This stemmed from two practical considerations. First, earlier layers predominantly process token-level information rather than the task-specific, prompt-agnostic features we aimed to identify. Second, our analysis revealed lower quality in IE approximations for these earlier layers, as detailed in Appendix E.2. Marks et al. (2024) similarly excludes the first third of the layers in their circuit analysis of Gemma 2 2B.

To evaluate the task specificity of discovered circuits, we conducted pairwise ablation studies examining cross-task faithfulness impacts. For each task circuit, we ablated the nodes with highest IEs until the task's faithfulness metric dropped to 0.3 — representing a substantial degradation of the model's original performance. We then measured how this ablation affected faithfulness scores across all other tasks. The results, presented in Figure 8, demonstrate that our extracted circuits exhibit strong task specificity, with performance degradation largely confined to their target tasks. We observed expected exceptions only between closely related task pairs, such as different translation tasks or conceptually similar tasks like *person_language* and *location_language*, where ablating one circuit naturally impacted performance on its counterpart.

## 4.2. Task-Detection Features

Our modified SFC analysis revealed a second crucial component of the ICL mechanism: task-detection features. Unlike executor features that activate before task completion, these features specifically activate when a task is completed in the training data - precisely on the output tokens from the Example 1. Figure 9 contains two examples of such features. Both task-detection and task-execution features showed high Indirect Effects (IEs) in the extracted sparse feature circuits,

with task-detection features connected to task execution features through attention output and transcoder nodes. We applied our task vector cleaning algorithm to extract task-detection features, identifying layer 11 as optimal for steering, preceding the layer 12 task-execution features. The details can be found in Appendix G. As with executor features, we present the steering heatmap in Figure 10 and the activation mass percentages in Table 2. We again see the task and token-type specificity of these features.

| iomeland of ties – Croatia. Th | his item by giving it a score |
| Reuters) - Bulgaria's presiden | about it by sending our help ( |
| he Dubline: Ireland's oldest | ipped. By doing this the few |

(a) Country detector feature 11459.

(b) Gerund form detector feature 8446.

*Figure 9.* A subset of max activating examples for detector features from Appendix I.

| Token Type | Mass (%) |
|---|---|
| output | 96.76 |
| input | 3.22 |
| newline | 0.01 |
| arrow | 0.0 |
| prompt | 0.0 |

*Table 2.* Activation masses for **task-detection** features across different token types, averaged across all tasks. We can notice that they activate almost exclusively on output tokens.

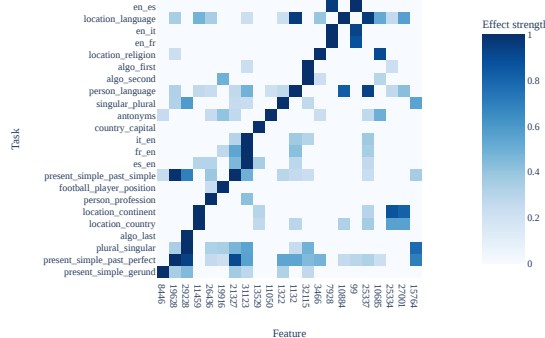

*Figure 10.* Heatmap showing the effect of steering with the task-detection feature most relevant to each task, on every task. We see that task detection features are typically specific to the task, with exceptions for similar tasks.

To evaluate the causal connection between task-detection features and task-execution features, we matched the strongest detection and execution features for each task based on their steering effects. We then ablated detection directions while fixing attention patterns and measured the de-

crease in execution features activations. Figure 11 presents the results.

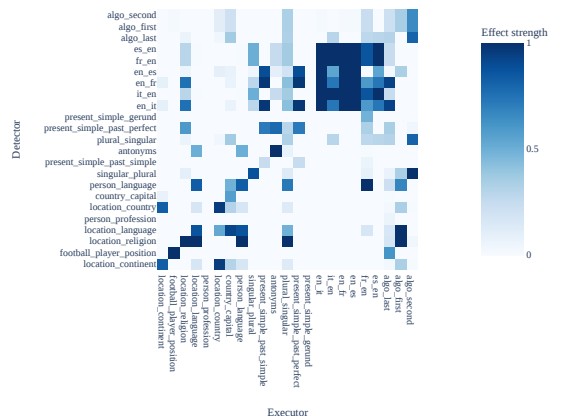

*Figure 11.* Heatmap showing the fraction of the executor activation lost of the top task-detection features of each task, after the detectors were ablated. Averaged across all initial non-zero activations in all tasks.

Our causal analysis demonstrated that ablating task-detection features substantially reduced the activation of their corresponding task-execution features, validating their hypothesized roles in the ICL circuit. Translation tasks exhibited particularly high interconnectivity, pointing to shared circuitry across this task family. Two tasks diverged from this pattern: *person_profession* and *present_simple_gerund* showed unusually weak detection-execution connections, suggesting a need for deeper investigation.

## 5. Related work

**Mechanistic Interpretability**   Olah et al. (2020) defines a framing for mechanistic interpretability in terms of *features* and *circuits*. It claims that neural network latent spaces have directions in them called features that correspond to meaningful variables. These features interact through model components sparsely to form circuits: interpretable computation subgraphs relevant to particular tasks. These circuits can be found through manual inspection in vision models Cammarata et al. (2020). In language models, they can be found through manual patching (Wang et al., 2022; Hanna et al., 2023; Lieberum et al., 2023; Chan et al., 2022) or automated circuit discovery (Conmy et al. (2023); Syed et al. (2023); Bhaskar et al. (2024), though see Miller et al. (2024)). Marks et al. (2024) extends this research area to use **Sparse Autoencoders**, as discussed below.

**In-Context Learning (ICL)**   ICL was first introduced in Brown et al. (2020) and refers to models learning to perform tasks from prompt information at test time. There is a large

area of research studying its applications Dong et al. (2024), high-level mechanisms Min et al. (2022) and limitations Peng et al. (2023). Elhage et al. (2021) and Olsson et al. (2022) find *induction heads* partly responsible for in-context learning. However, since these attention heads do more than just induction Goldowsky-Dill et al. (2023), and are not sufficient for complex task-following, induction heads alone cannot explain ICL. Anil et al. (2024, Appendix G) proposes a mechanistic hypothesis for an aspect of simple in-context task behavior. (Hendel et al., 2023) and (Todd et al., 2024) find that simple in-context learning tasks create strong directions in the residual stream adding which makes it possible for a network to perform tasks zero-shot, but does not explain how task vectors form nor what interpretable components the task vectors are composed of. A more detailed discussion can be found in Appendix H. Of particular interest is (Wang et al.), which investigates a simple ICL classification task and finds similar results with different terminology (information flow instead of circuits, "label words" instead of task-detection features). Recent work by (Park et al., 2024) demonstrates that language models reorganize existing representations of objects to adapt them to new tasks from the current context. This finding suggests that for the tasks they study, our task executing features may become more detached from their maximum activating examples, presenting an interesting class of tasks for applying our methods.

**Sparse Autoencoders**   Superposition, where interpretable neural network units don't align with basis directions, remains a key challenge in mechanistic interpretability Elhage et al. (2022). Sparse autoencoders (SAEs) Ng (2011); Bricken et al. (2023) address this, with recent works improving their training Rajamanoharan et al. (2024b); Bussmann et al. (2024); Braun et al. (2024). Building on this, Cunningham et al. (2023) applies automated circuit discovery to small language models, while Marks et al. (2024) adapts attribution methods in the SAE basis. Dunefsky et al. (2024) introduces transcoders to simplify MLP circuit analysis, which we incorporate into our Gemma 1 SAE suite.

## 6. Conclusion

We use sparse autoencoders (SAEs) to explain in-context learning with unprecedented mechanistic detail. Our work demonstrates that SAEs serve as valuable circuit analysis tools, with our key innovations including TVC (Section 3.1) and SFC adaption for ICL (Section 2.2). We will also plan to share SAE training codebase in JAX with a full suite of SAEs for Gemma 1 2B after the paper publication. These advances lay the groundwork for analyzing more complex tasks and larger models.

**Limitations** Our analysis focused on the simple task vector setting to study in-context learning (Section 2.3), which represents only a subset of ICL applications in practice. While our SFC analysis centered on Gemma 1 2B, we successfully identified task execution features across multiple model architectures and scales, supporting the broader applicability of our findings. This aligns with prior work showing that task vectors exist across models Todd et al. (2024).

In our analysis, we often saw multiple competing execution and detection features, though these features remained highly task-specific in nature — a pattern that appears common in LLM interpretability due to their regenerative capabilities.

## Acknowledgements

This work was conducted during the ML Alignment & Theory Scholars (MATS) Program, which is sponsored by Open-Philanthropy. We are grateful to MATS for providing a collaborative and supportive research environment. We also thank the TPU Research Cloud (TRC) for providing computational resources. Special thanks to Matthew Wearden and McKenna Fitzgerald for their invaluable guidance and management throughout the project. We also extend our gratitude to the MATS and Lighthaven staff, as well as all contributors who provided insightful feedback and discussions that shaped this work.

## Impact Statement

This paper presents work whose goal is to advance the field of mechanistic interpretability, rather than machine learning in general. Whilst there are potential societal consequences to the advancement of the ML field, we feel that research that improves understanding of ML models is unlikely to additionally contribute to these consequences, rather giving the field more tools to avert them.

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

# A. Model and dataset details

For our experiments, we utilized the Gemma 1 2B model, a member of the Gemma family of open models based on Google's Gemini models Team et al. (2024). The model's architecture is largely the same as that of Llama Dubey et al. (2024) except for tied input and output embeddings and a different activation function for MLP layers, so we could reuse our infrastructure for loading Llama models. We train residual and attention output SAEs as well as transcoders for layers 1-18 of the model on FineWeb Penedo et al. (2024).

Our dataset for circuit finding is primarily derived from the function vectors paper Todd et al. (2024), which provides a diverse set of tasks for evaluating the existence and properties of function vectors in language models. We supplemented this dataset with three additional algorithmic tasks to broaden the scope of our analysis:

- Extract the first element from an array of length 4

- Extract the second element from an array of length 4

- Extract the last element from an array of length 4

The complete list of tasks used in our experiments with task descriptions is as follows:

| Task ID | Description |
|---|---|
| location_continent | Name the continent where the given landmark is located. |
| football_player_position | Identify the position of a given football player. |
| location_religion | Name the predominant religion in a given location. |
| location_language | State the primary language spoken in a given location. |
| person_profession | Identify the profession of a given person. |
| location_country | Name the country where a given location is situated. |
| country_capital | Provide the capital city of a given country. |
| person_language | Identify the primary language spoken by a given person. |
| singular_plural | Convert a singular noun to its plural form. |
| present_simple_past_simple | Change a verb from present simple to past simple tense. |
| antonyms | Provide the antonym of a given word. |
| plural_singular | Convert a plural noun to its singular form. |
| present_simple_past_perfect | Change a verb from present simple to past perfect tense. |
| present_simple_gerund | Convert a verb from present simple to gerund form. |
| en_it | Translate a word from English to Italian. |
| it_en | Translate a word from Italian to English. |
| en_fr | Translate a word from English to French. |
| en_es | Translate a word from English to Spanish. |
| fr_en | Translate a word from French to English. |
| es_en | Translate a word from Spanish to English. |
| algo_last | Extract the last element from an array of length 4. |
| algo_first | Extract the first element from an array of length 4. |
| algo_second | Extract the second element from an array of length 4. |

This diverse set of tasks covers a wide range of linguistic and cognitive abilities, including geographic knowledge, language translation, grammatical transformations, and simple algorithmic operations. By using this comprehensive task set, we aimed to thoroughly investigate the in-context learning capabilities of the Gemma 1 2B model across various domains.

# B. SAE Training

Our Gemma 1 2B SAEs are trained with a learning rate of 1e-3 and Adam betas of 0.0 and 0.99 for 150M ($\pm100$) tokens of FineWeb Penedo et al. (2024). The methodology is overall similar to Bloom (2024). We initialize encoder weights orthogonally and set decoder weights to their transpose. We initialize decoder biases to 0. We use (Rajamanoharan, 2024)'s ghost gradients variant (ghost gradients applied to dead features only, loss multiplied by the proportion of death features) with the additional modification of using softplus instead of exp for numerical stability. A feature is considered dead when

its density (according to a 1000-batch buffer) is below 5e-6 or when it has not fired in 2000 steps. We use Anthropic's input normalization and sparsity loss for Gemma 1 2B Conerly et al. (2024). We found it to improve Gated SAE training stability. We modified it to work with transcoders by keeping track of input and output norms separately and predicting normed outputs.

We convert our Gated SAEs into JumpReLU SAEs after training, implementing algorithms like TVC and SFC in a unified manner for all SAEs in this format (including simple SAEs). The conversion procedure involves setting thresholds to replicate the effect of the gating branch. For further details, see (Rajamanoharan et al., 2024b).

We use 4 v4 TPU chips running Jax Bradbury et al. (2018) (Equinox Kidger & Garcia (2021)) to train our SAEs. We found that training with Huggingface's Flax LM implementations was very slow. We reimplemented LLaMA Dubey et al. (2024) and Gemma Team et al. (2024) in Penzai Johnson (2024) with a custom layer-scan transformation and quantized inference kernels as well as support for loading from GGUF compressed model files. We process an average of around 4400 tokens per second, which makes training SAEs and not caching LM activations the main bottleneck. For this and other reasons, we don't do SAE sparsity coefficient sweeps to increase TPU utilization.

For caching, we use a distributed ring buffer which contains separate pointers on each device to allow for processing masked data. The (in-place) buffer update is in a separate JIT context. Batches are sampled randomly from the buffer for each training step.

We train our SAEs in bfloat16 precision. We found that keeping weights and scales in bfloat16 and biases in float32 performed best in terms of the number of dead features and led to a Pareto improvement over float32 SAEs.

For training Phi 3 Abdin et al. (2024) SAEs, we use data generated by the model unconditionally, similarly to Xu et al. (2024)[4]. The resulting dataset we train the model on contains many math problems and is formatted as a natural-seeming interaction between the user and the model.

Each SAE training run takes us about 3 hours. We trained 3 models (a residual SAE, an attention output SAE, and a transcoder) for each of the 18 layers of the model. This is about 1 week of v4-8 TPU time.

Our SAEs and training code will be made public after paper publication.

## C. Example circuits

An example output of our circuit cleaning algorithm can be found in Figure 12. We can see the flow of information through a single high-IE attention feature from a task-detection feature (activating on output tokens) to transcoder and residual execution features (activating on arrow tokens). The feature activates on antonyms on the detection feature #11050: one can assume the first sequence began as "Short Term Target", making the second half an antonym.

We will release a web interface for viewing maximum activating examples and task feature circuits.

## D. Task Vector Cleaning Algorithm

The task vector cleaning algorithm is a novel approach we developed to isolate task-relevant features from task vectors. Figure 13 provides an overview of this algorithm.

Our process begins with collecting residuals for task vectors using a batch of 16 and 16-shot prompts. We then calculate the SAE features for these task vectors. We explored two methods: (1) calculating feature activation and then averaging across tokens, and (2) averaging across tokens first and then calculating the task vector. They had similar performances.

The cleaning process is performed on a training batch of 24 pairs, with evaluation conducted on an additional 24 pairs. All prompts are zero-shot. An example prompt is as follows:

---

[4]Phi-3 is trained primarily with instruction following data, making it an aligned chat model.

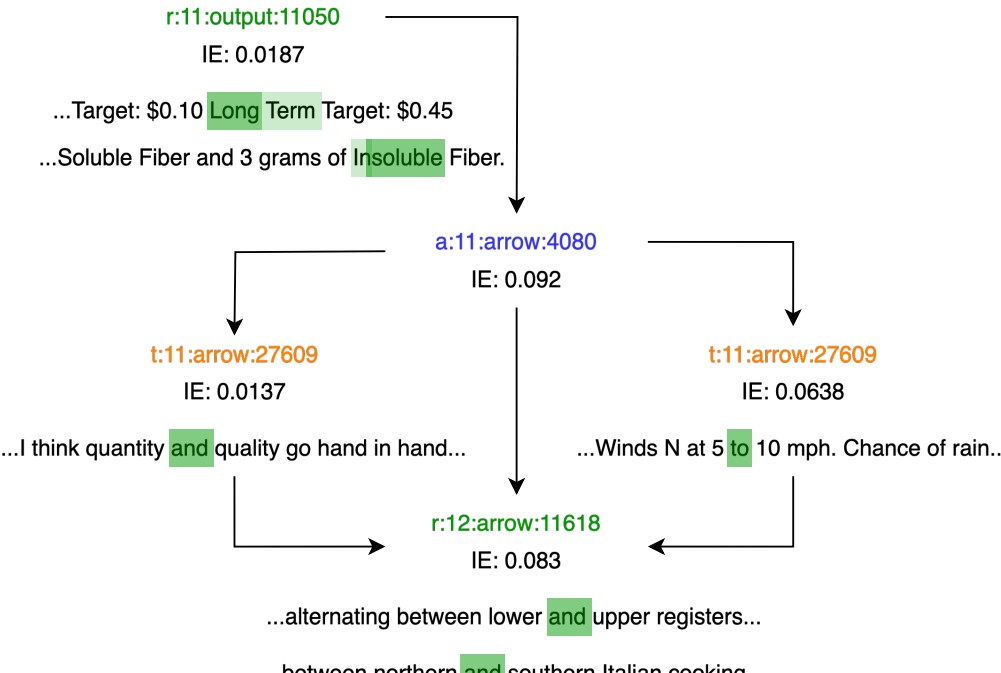

*Figure 12.* An example of a circuit found using our SFC variant. We focused on a subcircuit with high indirect effects. Maximum activating examples from the SAE training distribution are included.

| BOS | Follow | the | pattern | : | \n |
|-----|--------|-----|---------|---|-----|
| tall | → | short | \n | | |
| ... | | | | | |
| old | → | young | \n | | |
| hot | → | cold | | | |

*Example 2.* The steered token is highlighted in red. Loss is calculated on the yellow token.

The algorithm is initialized with the SAE reconstruction as a starting point. It then iteratively steers the model on the reconstruction layer and calculates the loss on the training pairs. To promote sparsity, we add the $L_1$ norm of weights with coefficient $\lambda$ to the loss function. The algorithm implements early stopping when the $L_0$ norm remains unchanged for $n$ iterations.

```python
def tvc_algorithm(task_vector, model, sae):
    initial_weights = sae.encode(task_vector)
    def tvc_loss(weights, tokens):
        task_vector = sae.decode(weights)
        mask = tokens == self.separator
        model.residual_stream[layer, mask] += task_vector
        # loss only on the ``output" tokens,
        # ignoring input and prompt tokens
        loss = logprobs(model.logits, tokens, ...)
        return loss + l1_coeff * l1_norm(weights)
    weights = initial_weights.copy()
```

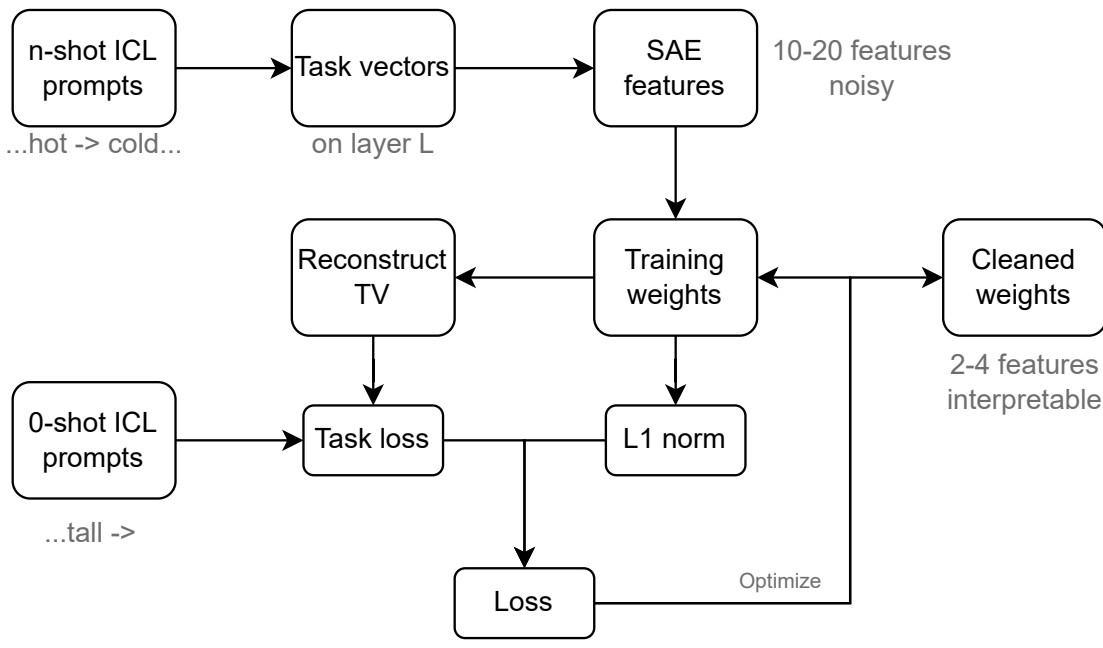

*Figure 13.* An overview of our Task Vector Cleaning algorithm. TV stands for Task Vector.

```
12    optimizer = adam(weights, lr=0.15)
13    last_l0, without_change = 0, 0   # early stopping
14    for _ in range(1000):
15      grad = jax.grad(tvc_loss)(weights, tokens)
16      weights = optimizer.step(grad)
17      if l0_norm(weights) != last_l0:
18        last_l0, without_change = l0_norm(weights), 0
19      elif without_change >= 50:
20        break
21    return weights
```

*Algorithm 1.* Pseudocode for Task Vector Cleaning.

The hyperparameters $\lambda$, $n$, and learning rate $\alpha$ can be fixed for a single model. We experimented with larger batch sizes but found that they did not significantly improve the quality of extracted features while substantially slowing down the algorithm due to gradient accumulation.

The algorithm takes varying amounts of time to complete for different tasks and models. For Gemma 1, it stops at 100-200 iterations, which is close to 40 seconds at 5 iterations per second.

It's worth noting that we successfully applied this method to the recently released Gemma 2 2B and 9B models using the Gemma Scope SAE suite Lieberum et al. (2024). It was also successful with the Phi-3 3B model Abdin et al. (2024) and with our SAEs, which were trained similarly to the Gemma 1 2B SAEs.

### D.1. $L_1$ Sweeps

To provide more details about the method's effectiveness across various models and SAE widths, we conducted $L_1$ coefficient sweeps with our Phi-3 and Gemma 1 2B SAEs, as well as Gemma Scope Gemma 2 SAEs. We chose two SAE widths for Gemma 2 2B and 9B: 16k and 65k. For Gemma 2 2B we also swept across several different target SAE $L_0$ norms. We studied only the optimal task vector layer for each model: 12 for Gemma 1, 16 for Gemma 2, 18 for Phi-3, and 20 for

Gemma 1

*Figure 14.* Performance of ITO and TVC across different tasks and optimization parameters compared to task vectors for Gemma 1 2B. The Y-axis shows relative improvement over task vector loss, while the X-axis shows the fraction of active TV features used. Metric calculation details are available in D.1

Gemma 2 9B. We used a learning rate of 0.15 with the Gemma 1 2B, Phi-3, and Gemma 2 2B 65k models, 0.3 with Gemma 2 2B 16k, and 0.05 with 200 early stopping steps for Gemma 2 9B.

Figures 14, 15, 16 compare TVC and ITO against original task vectors. The X-axis displays the fraction of active task vector SAE features used. The Y-axis displays the TV loss delta, calculated as $(L_{TV} - L_{Method})/L_{Zero}$, where $L_{TV}$ is the loss from steering with the task vector, $L_{Method}$ is the loss after it has been cleaned using the corresponding method, and $L_{Zero}$ is the uninformed (no-steering) model loss. This metric shows improvement over the task vector relative to the loss of the uninformed model. Points were collected from all tasks using 5 different $L_1$ coefficient values.

We observe that our method often improves task vector loss and can reduce the number of active features to one-third of those in the original task vector while maintaining relatively intact performance. In contrast, ITO rarely improves the task vector loss and is almost always outperformed by TVC.

Figures 17, 18 and 19 show task-mean loss decrease (relative to no steering loss) and remaining TV features fraction plotted against $L_1$ sweep coefficients. We see that $L_1$ coefficients between 0.001 and 0.025 result in relatively intact performance, while significantly reducing the amount of active SAE features. From Figure 18 we can notice that the method performs better with higher target l0 SAEs, being able to affect the loss with just a fraction of active SAE features.

## E. Details of our SFC implementation

### E.1. Implementation details

Our implementation of circuit finding attribution patching is specialized for Jax and Penzai.

We first perform a forward-backward pass on the set of prompts, collecting residuals and gradients from the metric to residuals. We collect gradients with `jax.grad` by introducing "dummy" zero-valued inputs to the metric computation function that are added to the residuals of each layer. Note that we do not use SAEs during this stage.

We then perform an SAE encoding step and find the nodes (residual, attention output, and transcoder SAE features and error nodes) with the highest indirect effects using manually computed gradients from the metric. After that, we find the features with the top K indirect effects for each layer and position mask and treat them as candidates for circuit edge targets. We compute gradients with respect to the metric to the values of those nodes, propagate them to "source features" up to one layer above, and multiply by the values of the source features. This way, we can compute indirect effects for circuit edges and prune the initially fully connected circuit. However, like (Marks et al., 2024), we do not perform full ablation of circuit edges.

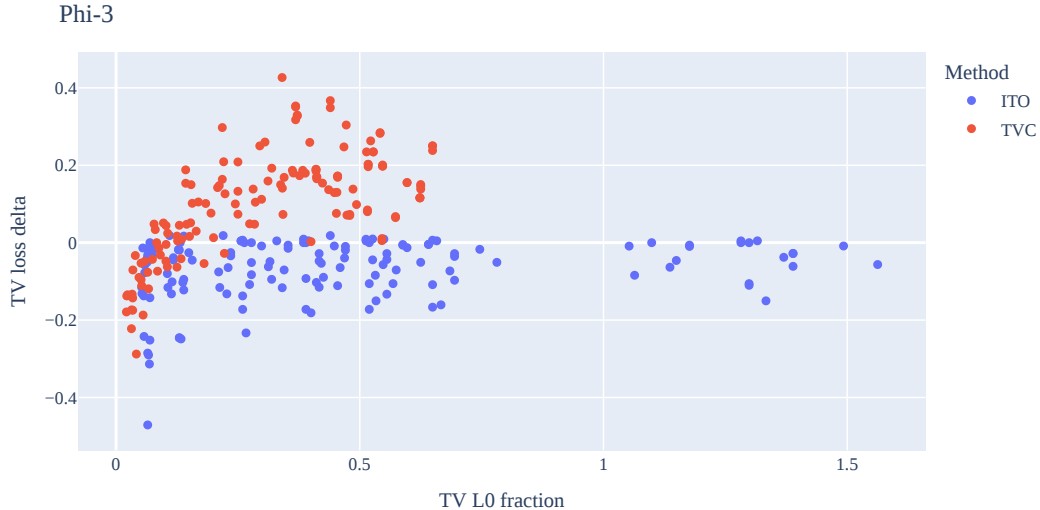

*Figure 15.* Performance of ITO and TVC across different tasks and optimization parameters compared to task vectors for Phi-3. The Y-axis shows relative improvement over task vector loss, while the X-axis shows the fraction of active TV features used. Metric calculation details are available in D.1

We include a simplified implementation of node-only SFC in Algorithm 2.

```
1   # resids_pre: L x N x D - the pre-residual stream at layer L
2   # resids_mid: L x N x D - the middle of the residual stream
3   # (between attention and MLP) at layer L
4   # grads_pre: L x N x D - gradients from the metric to resids_pre
5   # grads_mid: L x N x D - gradients from the metric to resids_mid
6   # all of the above are computed with a forward and backward
7   # pass without SAEs
8
9   # saes_resid: L - residual stream SAEs
10  # saes_attn: L - attention output SAEs
11  # transcoders_attn: L - transcoders predicting resids_pre[l+1]
12  # from resids_mid[l]
13
14  def indirect_effect_for_residual_node(layer):
15      sae_encoding = saes_resid[layer].encode(
16          resids_pre[layer])
17      grad_to_sae_latents = jax.vjp(
18          saes_resid[layer].decode,
19          sae_encoding
20      )(grads_pre[l])
21      return (grad_to_sae_latents * sae_encoding).sum(-1)
22
23  def indirect_effect_for_attention_node(layer):
24      sae_encoding = saes_attn[layer].encode(
25          resids_mid[layer] - resids_pre[layer])
26      grad_to_sae_latents = jax.vjp(
27          saes_attn[layer].decode,
28          sae_encoding
29      )(grads_mid[l])
30      return (grad_to_sae_latents * sae_encoding).sum(-1)
31
```

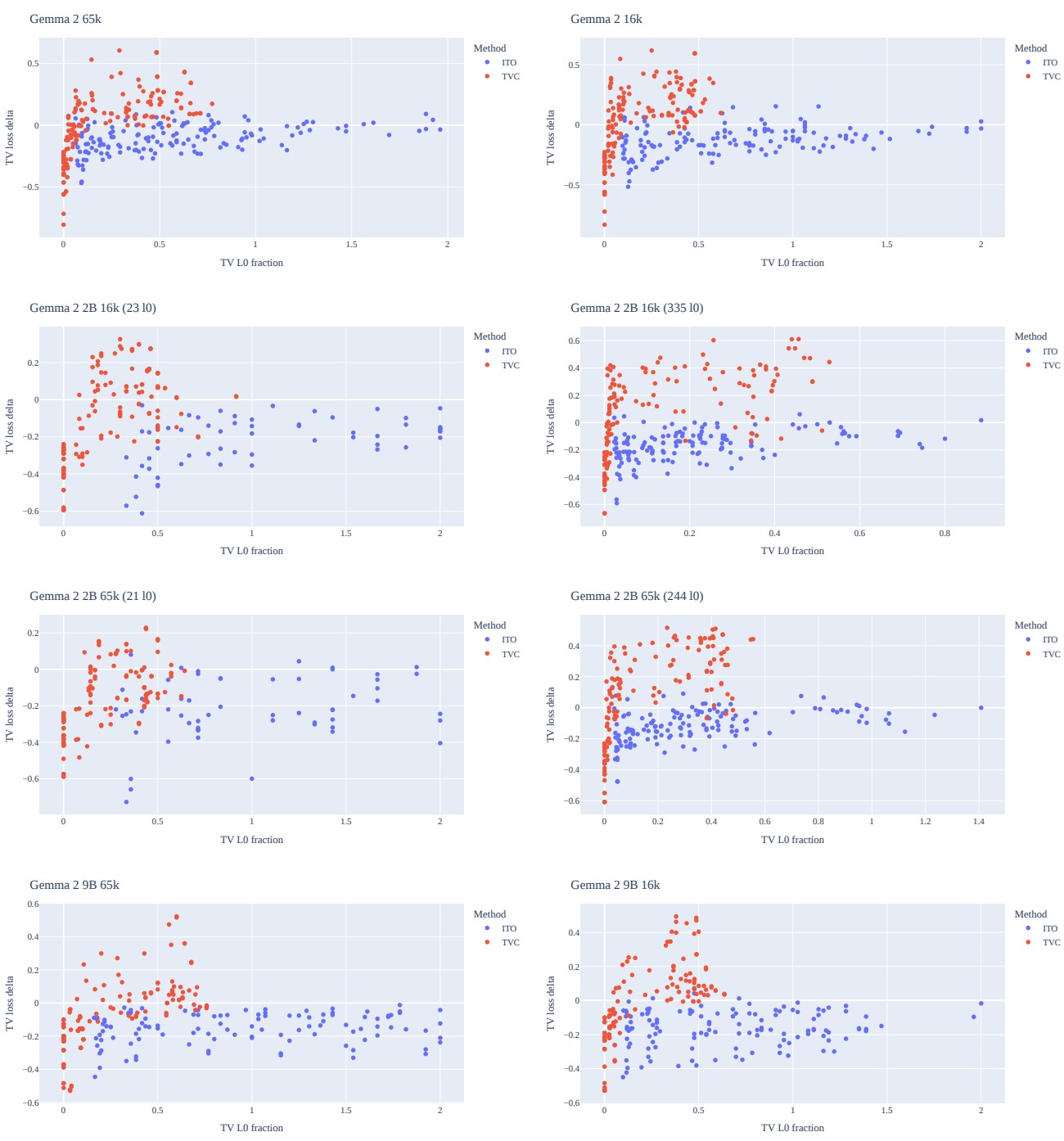

*Figure 16.* Performance of ITO and TVC across different tasks and optimization parameters compared to task vectors for Gemma 2 Gemma Scope SAEs. The Y-axis shows the relative improvement over the loss from steering with a task vector, while the X-axis shows the fraction of active TV features used. Metric calculation details are available in Appendix D.1.

```
32  def indirect_effect_for_transcoder_node(layer):
33      sae_encoding = transcoders[layer].encode(
34          resids_mid[layer])
35      grad_to_sae_latents = jax.vjp(
36          transcoders[layer].decode,
```

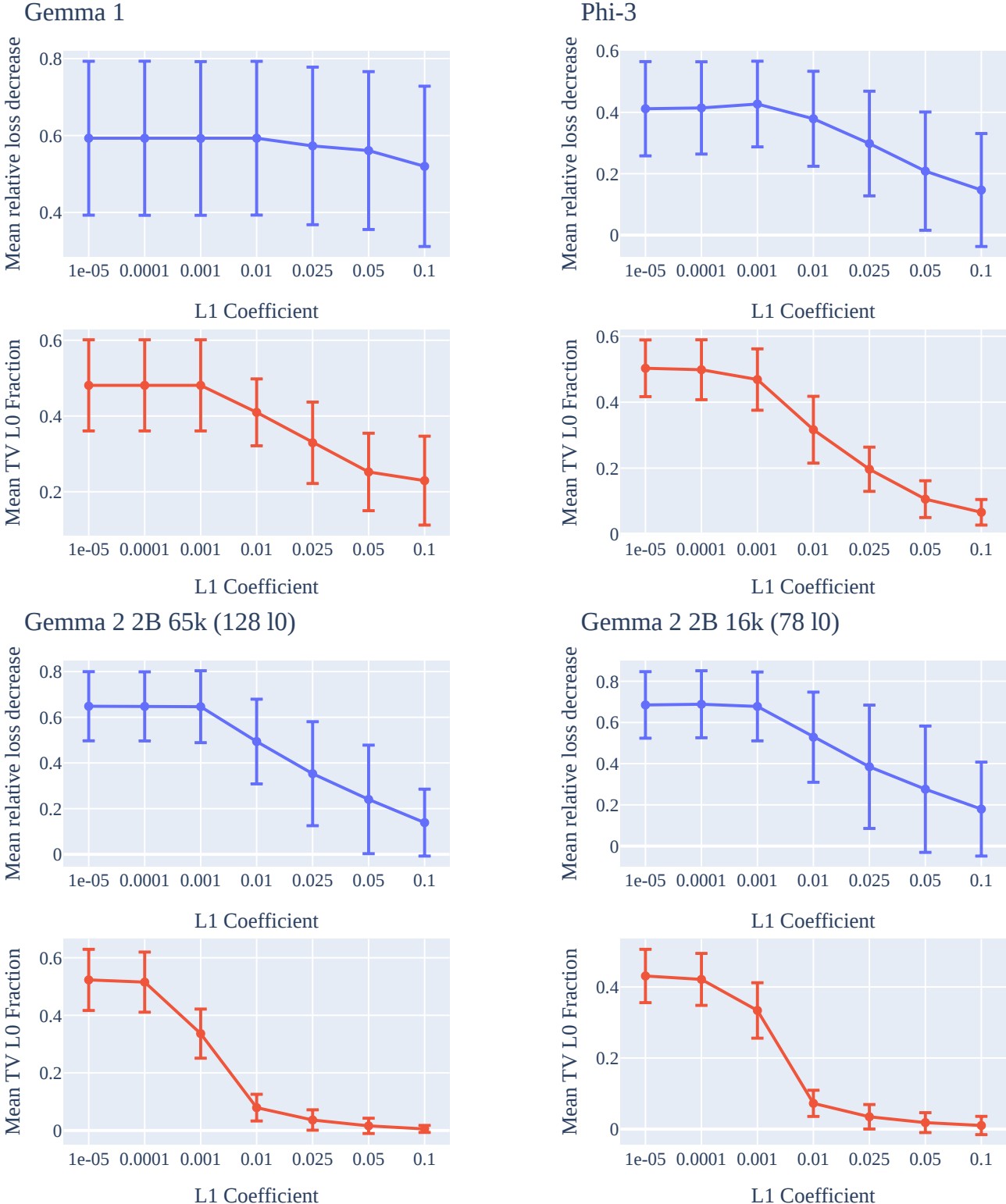

*Figure 17.* $L_1$ coefficient sweeps across different models and SAEs. All metrics are averaged across all tasks. Error bars show the standard deviation of the average for each case. Metric calculation details are available in D.1.

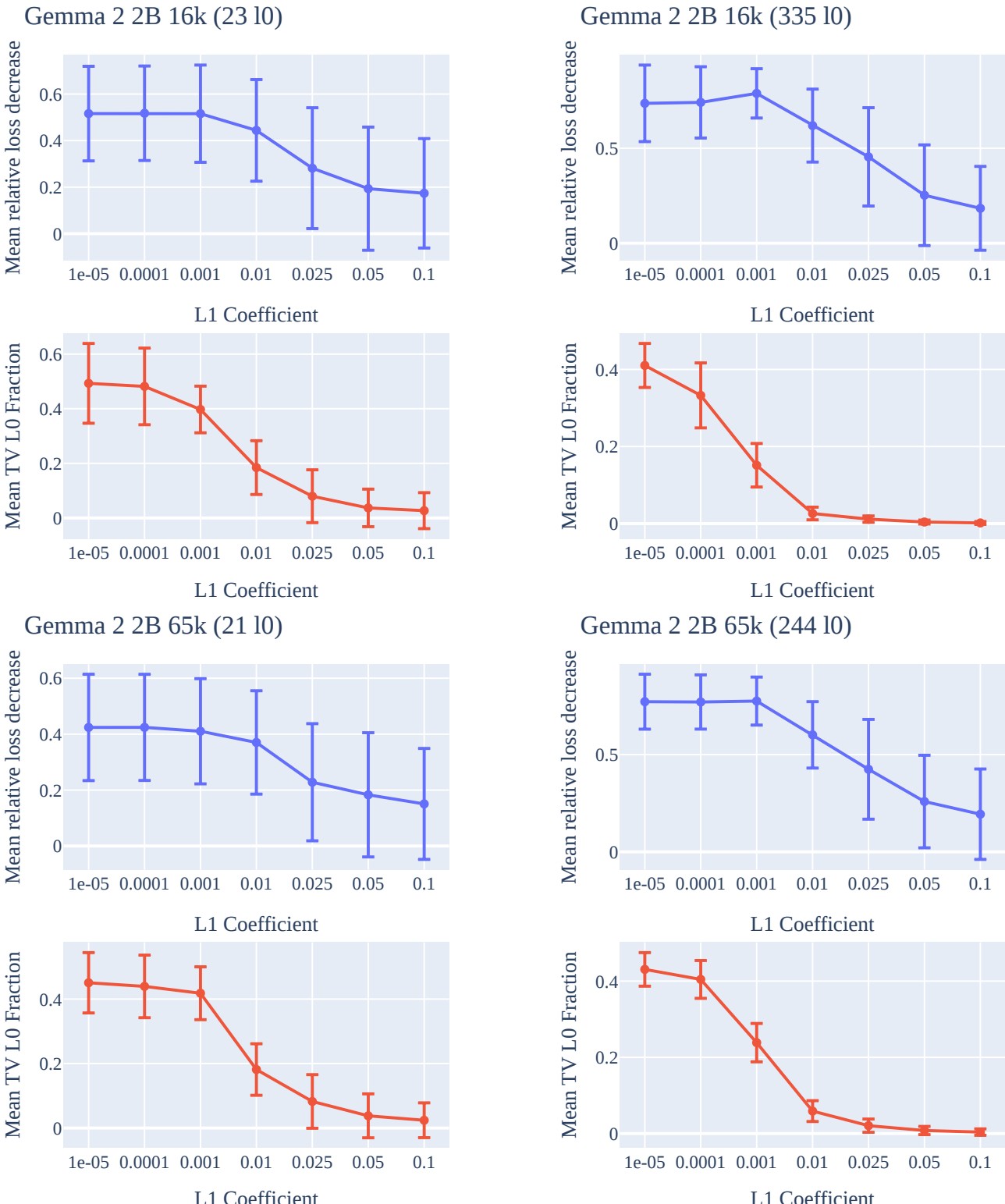

*Figure 18.* $L_1$ coefficient sweeps across different target SAE sparsities and widths for Gemma 2 2B. All metrics are averaged across all tasks. Error bars show the standard deviation of the average for each case. Metric calculation details are available in Appendix D.1.

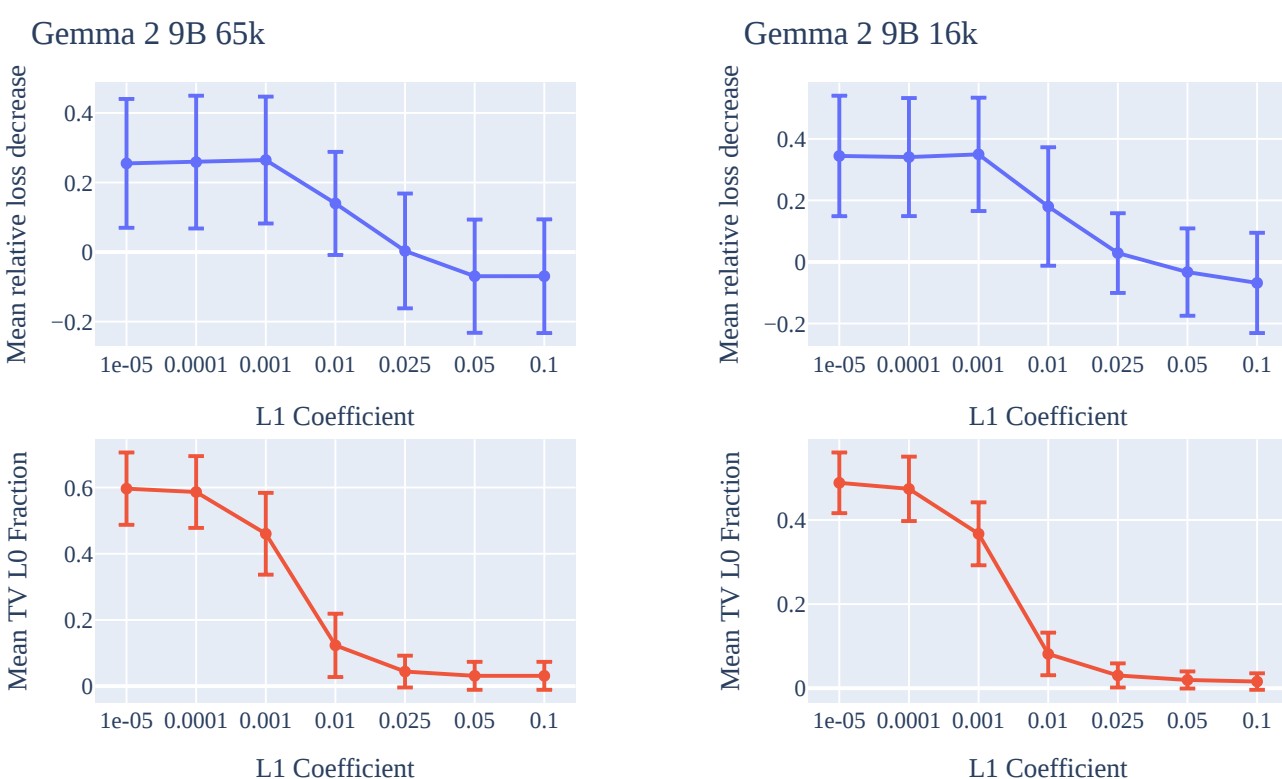

*Figure 19.* $L_1$ coefficient sweeps across two SAE widths for Gemma 2 9B. All metrics are averaged across all tasks. Error bars show the standard deviation of the average for each case. Metric calculation details are available in D.1.

```
37              sae_encoding
38          )(grads_pre[l+1])
39          return (grad_to_sae_latents * sae_encoding).sum(-1)
```

*Algorithm 2.* Pseudocode for Sparse Feature Circuits indirect effect calculation.

### E.2. IE approximation quality

Our IE calculation approach, which aggregates effects across all tokens of the same type, resulted in each layer having a limited number of non-zero nodes. This allowed us to directly examine the impact of disabling each of these nodes. We assessed the quality of the IE approximation by calculating correlation coefficients between the actual effects and their approximations. To further reduce computation time, we focused exclusively on nodes from the "input," "output," and "arrow" groups. Figure 20 displays the correlations averaged across all tasks for all SAE types combined, while Figure 21 presents the metric for each SAE type separately.

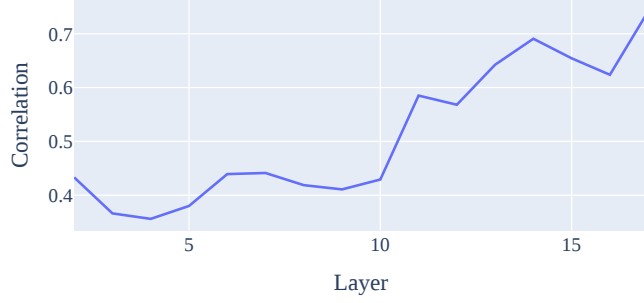

*Figure 20.* Average correlation of predicted and actual IEs across tasks for "input", "output" and "arrow" non-zero nodes.

Overall, we observe that the approximation quality remains relatively low before layer 6, which is much deeper in the model than layer 2, as reported by the original SFC paper. Non-residual stream SAEs begin to show adequate performance only in the last third of the model. This may be due to the quality of our trained SAEs, the increased task complexity, or token type-wise aggregation, and warrants further investigation. This is the primary reason our analysis focuses mainly on layers 10-15.

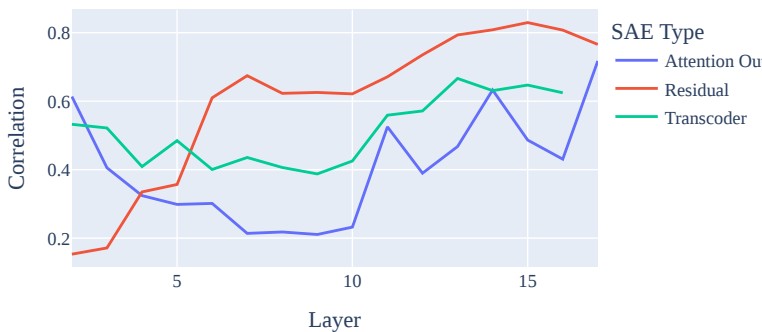

*Figure 21.* Average correlation of predicted and actual IEs across tasks for "input", "output" and "arrow" non-zero nodes for different SAE types.

## F. Steering with task-execution features

To evaluate the causal relevance of our identified ICL features, we conducted a series of steering experiments. Our methodology employed zero-shot prompts for task-execution features, measuring effects across a batch of 32 random pairs.

We set the target layer as 12 using Figure 2 and extracted all task-relevant features on it using our cleaning algorithm. To determine the optimal steering scale, we conducted preliminary experiments using manually identified task-execution features across all tasks. Through this process, we established an optimal steering scale of 15, which we then applied consistently across all subsequent experiments.

For each pair of tasks and features, we steered with the feature and measured the relative loss improvement compared to the model's task performance on a prompt without steering. This relative improvement metric allowed us to quantify the impact of each feature on task performance. Let $M(t, f) = L_{steered}/L_{zero}$, be the effect of steering on task $t$ with feature $f$.

To normalize our results and highlight the most significant effects, we applied several post-processing steps:

- $M(t, f) = \min(M(t, f), 1)$

- $M(t, f) = \left( M(t, f) - \min_{f'} M(t, f') \right) / \left( \max_{f'} M(t, f') - \min_{f'} M(t, f') \right)$

- $M(t, f) = 0$ if $M(t, f) < 0.2$

- Finally, we removed features with low maximum effect across all tasks to reduce the size of the resulting diagram. The full version of this diagram is present in Figure 22.

Prompt example with the steered token highlighted in red. Loss is calculated on the yellow token:

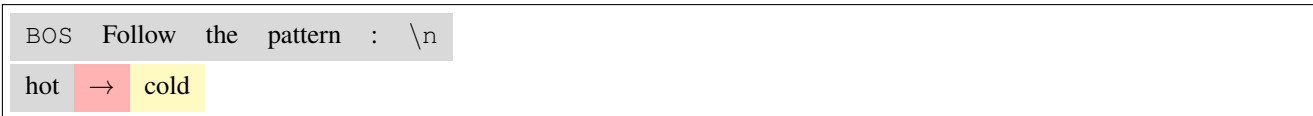

*Example 3.* Task-execution steering setup. The steered token is highlighted in red and the loss is calculated on the yellow token.

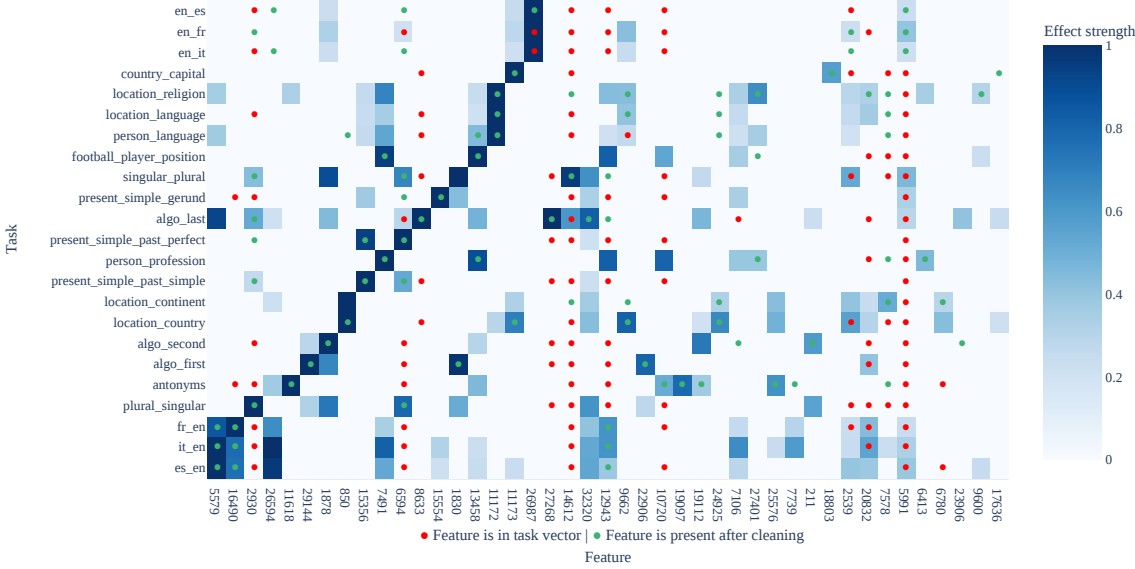

*Figure 22.* Full version of the heatmap in Figure 6 showing the effect of steering with individual task-execution features for each task. The features present in the task vector of the corresponding task are marked with dots (i.e. from the naive SAE reconstruction baseline in Section 3.1). Green dots show the features that were extracted by cleaning. Red dots are features present in the original task vector. Not all original features from the task vectors are present.

We also share the version of Figure 22 without normalization and value clipping. It is present in Figure 24. We see that task vectors generally contain just a few task-execution features that can boost the task themselves. The remaining features have much weaker and less specific effects.

## F.1. Negative steering

To further explore the effects of the executor feature, we also conducted negative steering experiments. The setup involved a batch of 16 ICL prompts, each containing 32 examples for each task. We collected all features from the cleaned task vectors for every task. Similar to positive steering, we steered with features on arrow tokens, but this time multiplying the direction by -1. Prompts this time contained several arrow tokens, and we steered on all of them simultaneously.

An important distinction from positive steering is that performance degradation in negative steering may occur due to two factors: (1) our causal intervention on the ICL circuit and (2) the steering scale being too high. To address this, we measured accuracy across all pairs in the batch instead of loss, as accuracy does not decrease indefinitely. We also observed that features no longer share a common optimal scale. Consequently, for each task pair, we iterated over several scales between 1 and 30. For each feature, we then selected a scale that reduced accuracy by at least 0.1 for at least one task. Steering results at this scale were used for this feature across all tasks.

Figure 23 displays the resulting heatmap. While we observe some degree of task specificity — and even note that some executing features from Figure 22 have their expected effects — we also find that negative steering exhibits significantly lower task specificity. Additionally, we observe that non-task-specific features have a substantial impact in this experiment. This suggests that steering experiments alone may not suffice for a comprehensive analysis of the ICL mechanism, thus reinforcing the importance of methods such as our modification of SFC.

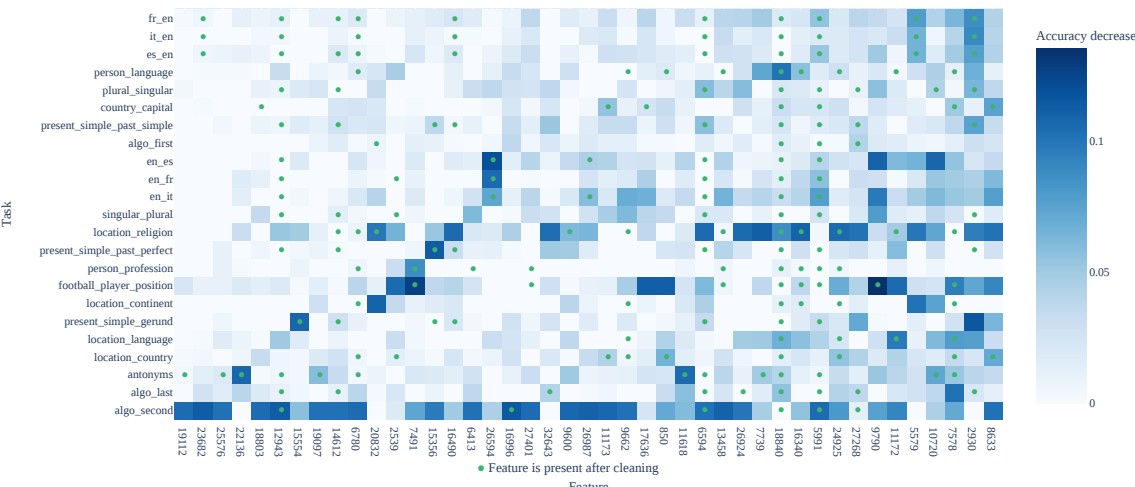

*Figure 23.* Negative steering heatmap. Displays accuracy decrease after optimal scale negative steering on full ICL prompts. Green circles show which features were present in the cleaned task vector of the corresponding task. More details in Appendix F.1.

## F.2. Gemma 2 2B positive steering

Additionally, we conducted zero-shot steering experiments with Gemma 2 2B 16k and 65k SAEs. Contrary to Gemma 1 2B, task executors from Gemma 2 2B did not have a single common optimal steering scale. Thus, we added an extra step to the experiment: for each feature and task pair, we performed steering with several scales from 30 to 300, and then selected the scale that had maximal loss decrease on any of the tasks. We then used this scale for this feature in application to all other tasks. Figure 25a and Figure 25b contain steering heatmaps for Gemma 2 2B 16k SAEs and Gemma 2 2B 65k SAEs respectively.

We observe a relatively similar level of executor task-specificity compared to Gemma 1. One notable difference between 16k and 65k SAEs is that 65k cleaned task vectors appear to contain more features with a strong effect on the task. However, this may be due to the $l_1$ regularization coefficient being too low.

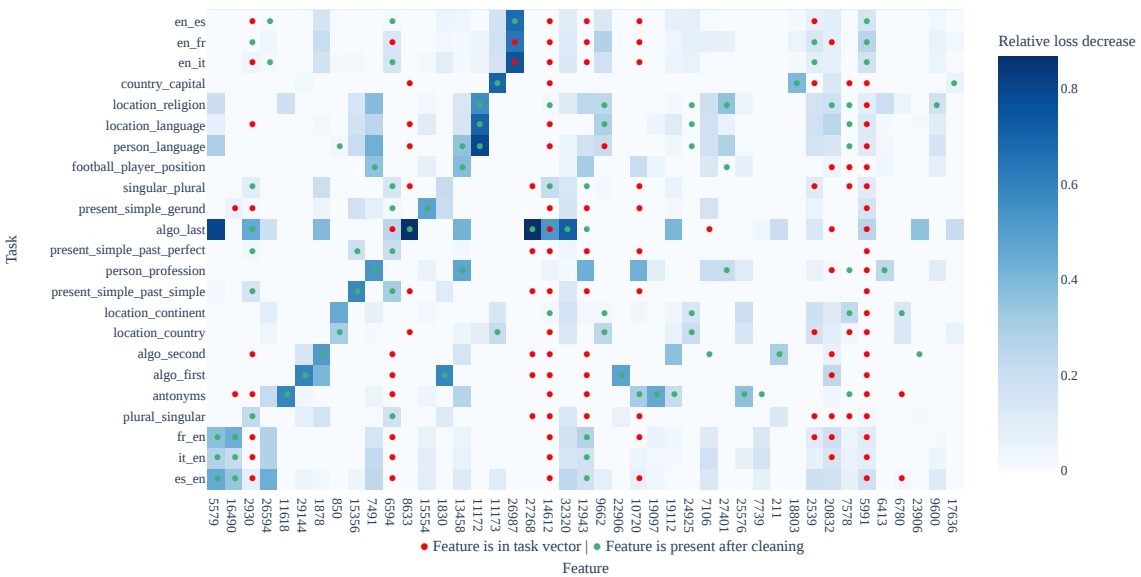

*Figure 24.* Unfiltered version of the heatmap in Figure 10 showing the effect of steering with individual task-execution features for each task. The features present in the task vector of the corresponding task are marked with dots. Green dots show the features that were extracted by cleaning. Red dots are the features present in the original task vector. Since the chart only contains features from cleaned task vectors, not all features from the original task vectors are present.

## G. Task-Detection Features

For our investigation of task-detection features, we employed a methodology similar to that used for task execution features, with a key modification. We introduced a fake pair to the prompt and focused our steering on its output. This approach allowed us to simulate the effect of the detection features the way it happens on real prompts.

Our analysis revealed that layers 10 and 11 were optimal for task detection, with performance notably declining in subsequent layers. We selected layer 11 for our primary analysis due to its proximity to layer 12, where we had previously identified the task execution features. This choice potentially facilitates a more direct examination of the interaction between detection and execution mechanisms.

The steering process for detection features followed the general methodology outlined in Appendix F, including the use of a batch of 32 random pairs, extraction of task-relevant features, and application of post-processing steps to normalize and highlight significant effects. The primary distinction lies in the application of the steering to the prompt.

This approach allowed us to create a comprehensive representation of the causal relationships between task-detection features and the model's ability to recognize specific tasks, as visualized in Figure 10.

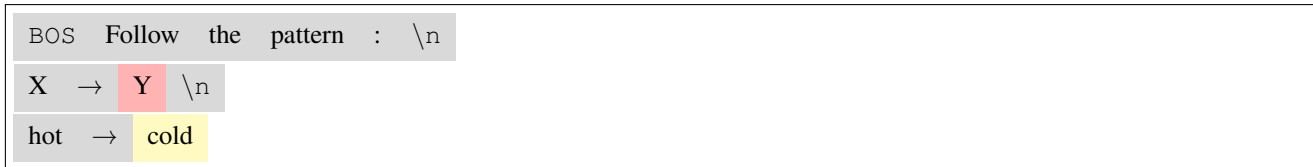

*Example 4.* Task-detection steering setup. The steered token is highlighted in red and the loss is calculated on the yellow token.

## H. ICL interpretability literature review

This section will cover work on understanding ICL not mentioned in Section 5.

(Raventós et al.) provides evidence for two different Bayesian algorithms being learned for linear regression ICL: one for limited task distributions and one that is similar to ridge regression. It also intriguingly shows that the two solutions lie in

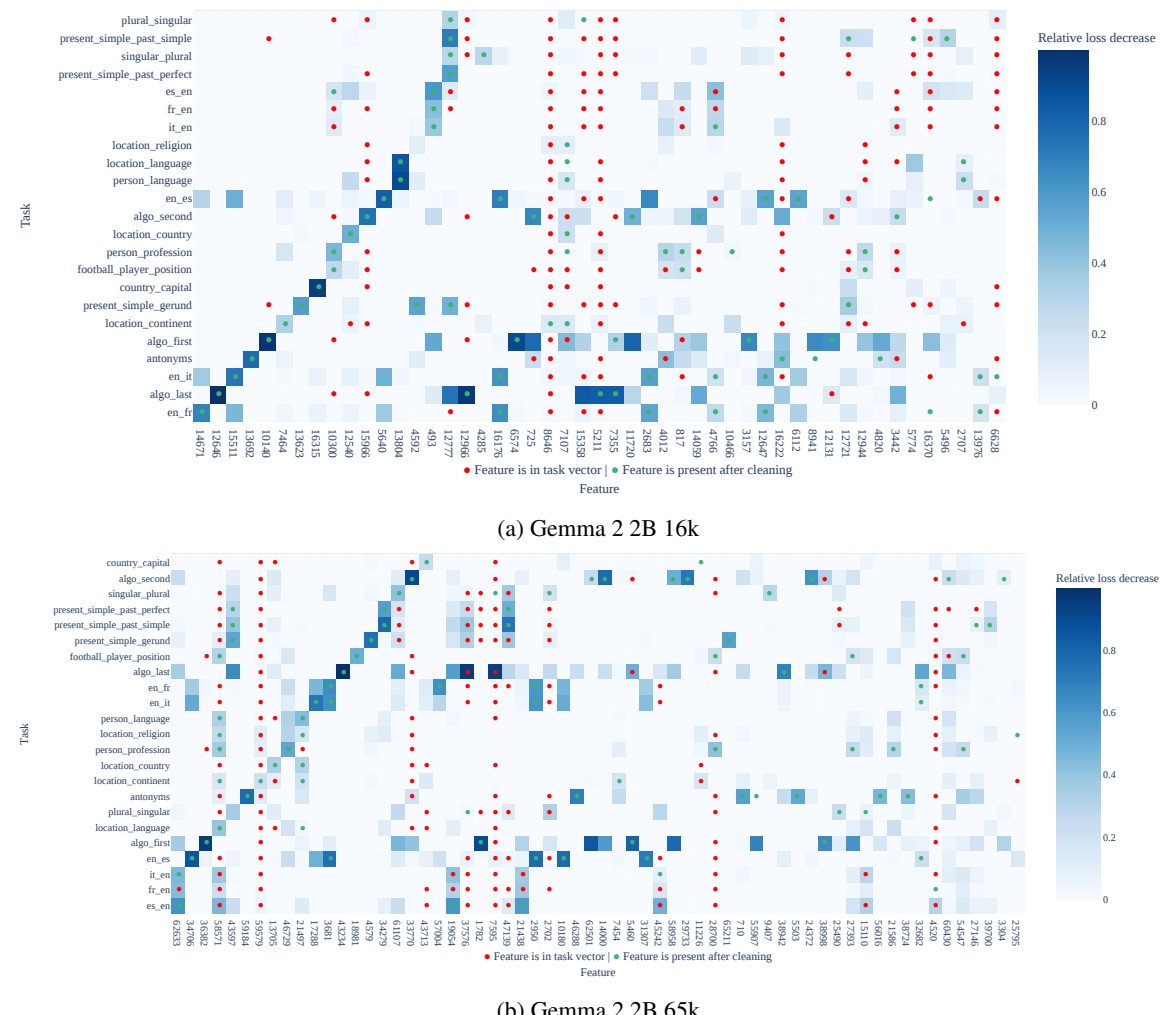

(a) Gemma 2 2B 16k

(b) Gemma 2 2B 65k

*Figure 25.* Unfiltered positive steering heatmap for Gemma 2 2B SAEs showing the effect of steering with individual task-execution features for each task. Steering scales were optimized for each feature. The features present in the task vector of the corresponding task are marked with dots. Green dots show the features that were extracted by cleaning. Red dots are the features present in the original task vector. Since the chart only contains features from cleaned task vectors, not all features from the original task vectors are present.

different basins of the loss landscape, a phase transition necessary to go from one to the other. While interesting, it is not clear if the results apply to real-world tasks.

The existence of discrete task detection and execution features hinges on the assumption that in-context learning works by classifying the task to perform and not by learning a task. (Pan et al.) aims to disentangle the two with a black-box approach that mixes up outputs to force the model to learn the task from scratch. (Si et al.) look at biases in task recognition in ambiguous examples through a black-box lens. We find more clear task features for some tasks than others but do not consider whether this is linked to how common a task is in pretraining data.

(Xie et al.) proposes that in-context learning happens because language models aim to model a latent topic variable to predict text with long-range coherence. (Wang et al., 2024) show following the two proposed steps rigorously improves results in real-world models. However, they do not endeavor to explain the behavior of non-finetuned models by looking at internal representations; instead, they aim to improve ICL performance.

(Han et al.) use a weight-space method to find examples in training data that promote in-context learning using a method akin to (Grosse et al., 2023), producing results similar to per-token loss analyses in (Olsson et al., 2022), and, similarly to the studies mentioned above, finds that those examples involve long-range coherence. Our method is also capable of finding

examples in data that are similar to ICL, and we find crisp examples for many tasks being performed Appendix I.

(Bansal et al.) offers a deeper look into induction heads, scaling up (Olsson et al., 2022) the way we scale up (Marks et al., 2024). Crucially, it finds that MLPs in later layers cannot be removed while preserving ICL performance, indirectly corroborating our findings from Section 4.2. (Chen et al.) come up with a proof that states that gradient flow converges to a generalized version of the algorithm suggested by (Olsson et al., 2022) when trained on n-gram Markov chain data.

(Garg et al.) studies the performance of toy models trained on in-context regression various *function classes*. (Yadlowsky et al.) find that Transformers trained on regression with multiple function classes have trouble combining solutions for learning those functions. (Oswald et al.) construct a set of weights for linear attention Transformers that reproduce updates from gradient descent and find evidence for the algorithm being represented on real models trained on toy tasks. (Mahankali et al.) proves that this algorithm is optimal for single-layer transformers on noisy linear regression data. (Shen et al.) questions the applicability of this model to real-world transformers. (Bai et al.) finds that transformers can switch between multiple different learning algorithms for ICL. (Dai et al.) find multiple similarities between changes made to model predictions from in-context learning and weight finetuning.

While important, we do not consider this direction of interpreting transformers trained on regression for concrete function classes through primarily white-box techniques. Instead, we aim to focus on clear discrete tasks which are likely to have individual features.

The results of (Wang et al.) are perhaps the most similar to our findings. The study finds "anchor tokens" responsible for aggregating semantic information, analogous to our "output tokens" (Section 2.3) and task-detection features. They tackle the full circuit responsible for ICL bottom-up and intervene on models using their understanding, improving accuracy. Like this paper, they do not deeply investigate later attention and MLP layers. Our study uses SAE features to find strong linear directions on output and arrow tokens corresponding to task detection and execution respectively, offering a different perspective. Additionally, we consider over 20 diverse token-to-token tasks, as opposed to the 4 text classification datasets considered in (Wang et al.).

# I. Max Activating Examples

This section contains max activating examples for some executor and detector features for Gemma 1 2B, as described in Bricken et al. (2023). They are computed by iterating over the training data distribution (FineWeb) and sampling activations of SAE features that fall within disjoint buckets for the activation value of span 0.5. We can observe that the degree of intuitive interpretability depends on the amount of task-similar contexts in the training data and SAE width.

We also provide max activating examples for Gemma 2 2B executor features from Figures 25b and 25a. These max activating examples are taken from the Neuronpedia Lin (2023) and are available in Figures 29 and 28.

Here we can notice the main difference between executors and detectors: executors mainly activate **before the task completion**, while detectors activate on the **token that completes the task**. We also found that in Gemma 1 2B detector features for some tasks were split between several token-level features (like the journalism feature in Figure 27f), and they did not create a single feature before the task executing features activated. We attribute this to the limited expressivity of the SAEs that we used.

st by alternating between lower and upper registe

erences between northern and southern Italian co

l models, both import and domestic, Ulmer's spec

between fresh and traditional, casual and elegant,

ces and both local and remote event logging. That

globally in both tropical and temperate waters. Blu

e, light and darkness, life and death. This assembl

(a) Max activating examples for the antonyms executor feature 11618.

Judgment Staff (裁きの杖, Sabaki no Tsue?), also known

Rift The Judgment Staff (裁きの杖, Sabaki no Tsue?), also

more commonly known as the Four-Tails (四尾, Yonbi), is a

as the Four-Tails (四尾, Yonbi), is a tailed beast sealed

1 meters dybde er vigtige for et-årige afgr

te vinden (inclusief een directe link naar de publicatie online als dez

elders te vinden (inclusief een directe link naar de publicatie online

een publicatie elders te vinden (inclusief een

Oh (侍合体シンケンオー Samurai Gattai Shinken'ō?)

দোর গোয়েন্দোগিরি

naar de publicatie online als deze beschikbaar is in een

atie online als deze beschikbaar is in een database op het internet).

Goendagiri (ফল‍ুদ‍োর গোয়েন‍্দাগ

did, upstage<bos>Israel (ישראל נתדים) is a small yet diverse

Agencia Española de Medicamentos y Productos Sanitarios, A

authority (Agencia Española de Medicamentos y Productos Sanitar

(b) Max activating examples for the English to foreign language translation executor feature 26987.

cultural diversity and that special joie de vivre (joy of life), that Mont

of creating Papel Picado Banderitas (little paper banners). Popular t

nicknames: "la ciudad dorada" (the golden city). Salamanca is also t

from the same root as jihad, or struggle, in the sense that ijti

conurbation "tsukin jigoku," or commuter hell. Images of rail worker:

, he acted according to our Sunna (tradition), and whoever slaughter

, and, of course, your karma (good and bad). John brings a wealth

The "it-sa Sicherheitsmesse" (security trade show), OWASP conferer

Cervesería Catalan along with a caña (draft beer) and a rosé...yes

s Apostle! I slaughtered the Nusuk (before the prayer) but I<bos>Tru

the living entities; sva-artha—interest; vyatikramah—ob

by her given name (Angelella = little angel), but called her Columba

(c) Max activating examples for the translation to English executor feature 5579.

working, connecting with diverse people and seeking out sustain

croll, and 2) Isolating unifying elements that transcend the indivi

nt like landing on the moon or the discovery of DNA. The focus

by using our search feature or by following the links above. Feel

as Liking and Favoriting photos, but it will expire after

spends her free time traveling and visiting exotic locations arou

enses tighten, grabbing offensive rebounds and making putback

er than participating in or observing or<bos>I tend to specialise i

nother, rather than participating in or observing or<bos>I tend to

a passenger car with plastics sheets and inhaling toxic fumes fr

nilies when going a long distance or flying with them when we ca

(d) Max activating examples for the "next comes gerund form" executor feature 15554.

on came all the way from Oslo Norway for the event. This has

migrated to New York City from the Galicia area, in northwest Spain.

nal. There were a few folks from Canada (British Columbia, Ontario and Quebec

an immigrant from Bangladesh who was granted political asylum by the

on and world number six Li Na of China. Azarenka, who is

hood in Seattle to my college years in Boston,

owever, batteries from rival manufacturers in the U.S. are exempt from

e USA and four in England. Of these, only three have

(e) Max activating examples for the prediction of city/country feature 850.

scientistb, - Rury Holman, directorc on behalf of the United Kingdom

Stinton, NAR CEO Charlie Young, President/CEO

, San Fernando Realty Dale Stinton, NAR CEO Charlie Young, President/

director ()a, - Philip Clarke, research fellowa, - Andrew Farmer

Hummel, MD, Ezio Bonifacio, PHD, and Anette-G.

lives in Bombay. Faruq Hassan: Poet and critic; teaches at Dawson Coll

<bos>lila Williams, President Randall Ramsay, Vice President Texas Cha

(f) Max activating examples for the person's occupation executor feature 13458.

*Figure 26.* Max activating examples for executor features from Figure 6.

Target: $0.10 Long Term Target: $0.45

Soluble Fiber and 3 grams of Insoluble Fiber. Ground Flaxseeds are a go

5 In. x 6 In.; Outer Dimensions: 7 In. x

a service: |Morning Services||Evening Service| Morning Worship at 8

temp: 15°C min temp: 11°C

Upper Zone and 75 Bottles in Lower Zone - Read More... The

page and 30% viewing the right half" "apple's decision

integration is performed first, followed by the quantitative combination.

access to the content item. Returns FALSE if the current<bos>|Oracle®

. As we alternate between defensive positions and offensive positions, w

other system that substantively uses<bos>Wikipedia sobre física de partículas

Directed By Tom Grundy Es gibt noch keine Kommentare. Sei der erste ...<bos>

006 - 213 halaman The book was selected as one of

Hour | Webcast - enregistré | Où et quand What is the Webcast About

[score hidden] 23 Minuten zuvor You just said 'if you exposed

hazard [score hidden] 23 Minuten zuvor You just said 'if you

vA-LINKER biedt mogelijkheden om een publicatie elders te vinden (

(a) Max activating examples for the antonyms detector feature 11050.

(b) Max activating examples for the English to foreign language switch detector feature 7928.

Say You can rate this item by giving it a score of one (poor),

, please let us know about it by sending our help desk an email .

which your order will be shipped. By doing this the few products

<pad><pad><pad><bos>Search for music by typing a word or ph

a one month non-recurring subscription by sending a cashier's c

s... Learn more about Concordia by following the links below: Co

this product deliver? Pay it forward by sharing what you loved (a

. Browse: Browse the database by applying one or more filters to

we are celebrating Valentine's Day by sharing some gorgeous a

page needs content. You can help by adding a sentence or a ph

NewsOK. He composed the ad by animating still photos taken b

Superficie Lunare (Composizione)" (Lunar surface - composition), execut

hardline group Tawhid wal Jihad (Monotheism and Holy War).

hardline group Tawhid wal Jihad (Monotheism and Holy War). One

hid wal Jihad (Monotheism and Holy War). One civilian was among

line group Tawhid wal Jihad (Monotheism and Holy War). One civilian

that reads "Arbeit Macht Frei" ("Work Brings Freedom") is a seminal mor

Tavola di San Giuseppe (St. Joseph's Feast). You'll

As part of the Tres Fronteras (Three Borders) area that includes Foz and

(c) Max activating examples for the gerund form detector feature 8446.

(d) Max activating examples for the translation to English detector feature 31123.

the homeland of ties – Croatia. There we found three local brands that

FIA (Reuters) - Bulgaria's president on Thursday called for a

s>Welcome to The Dubline: Ireland's oldest and newest discovery trail.

d><pad><pad><pad><pad><pad><bos>BulgariaSki.com is owned and ma

should know that "Deutschland" means Germany in German. Germany is

urban Budapest sketch... (Hungary) The old building standing on V

s>Message Behind African Heaters For Norway Spoof An online video, u

More of a Switzerland: More Personal Ads from the London Review

><bos>I have been a technology journalist and consultant for near

donation will help independent Adventist journalism expand across

an 50 journalists gathered at Klosters, a Swiss ski

tting punters, journalists, football managers and players. We also

Modelo<bos>The award-winning journalist Robert Fisk gave the in

Pulitzer Prize-winning journalist, formerly with The Washington Po

ew. If you are a journalist seeking comment on a story or more info

<bos>Peripatetic journalist and translator Porter (Road to Heaven:

n will likely endanger the lives of journalists and aid workers in the

houghtful post about the hazards of journalism following revelatio

about interviewing and journalism. Just like a marketing person do

(e) Max activating examples for the country detector feature 11459.

(f) Max activating examples for the journalist feature 26436. (The strongest detector for the person_profession task).

*Figure 27.* Max activating examples for detector features from Figure 10.

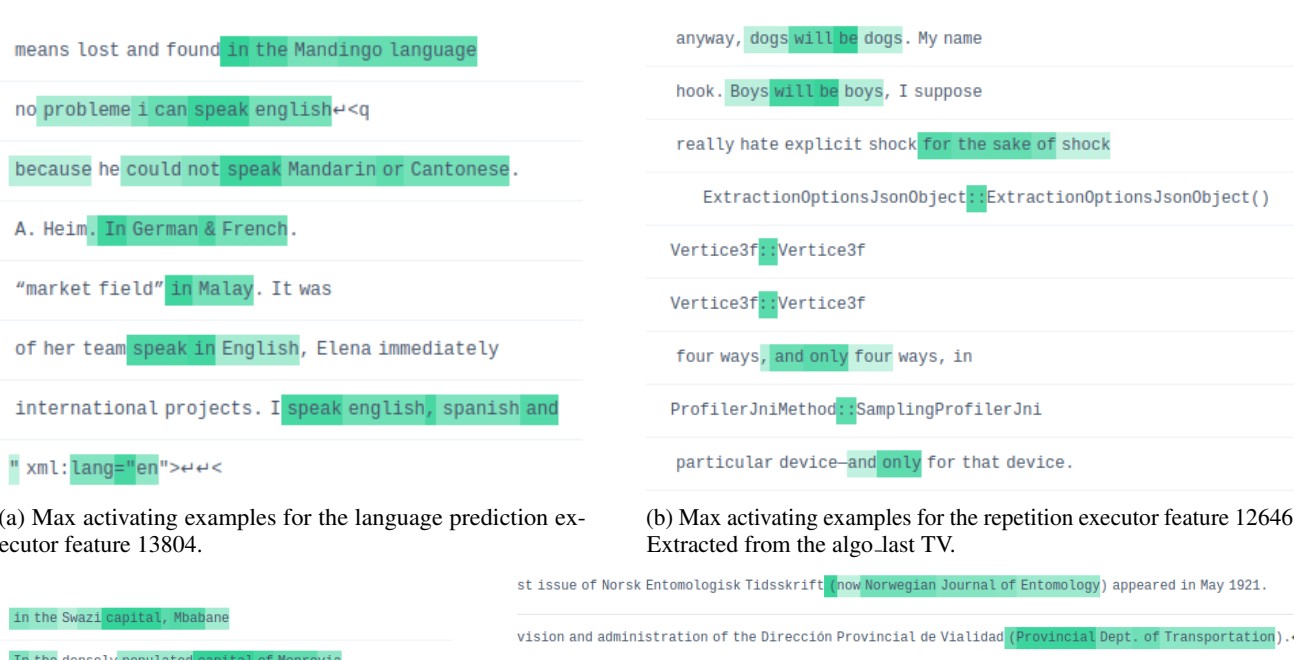

(a) Max activating examples for the language prediction executor feature 13804.

(b) Max activating examples for the repetition executor feature 12646. Extracted from the algo_last TV.

(c) Max activating examples for the capital prediction executor feature 16315.

(d) Max activating examples for the translation feature 493.

Figure 28. Max activating examples for Gemma 2 2B 16k executor features from Figure 25a.

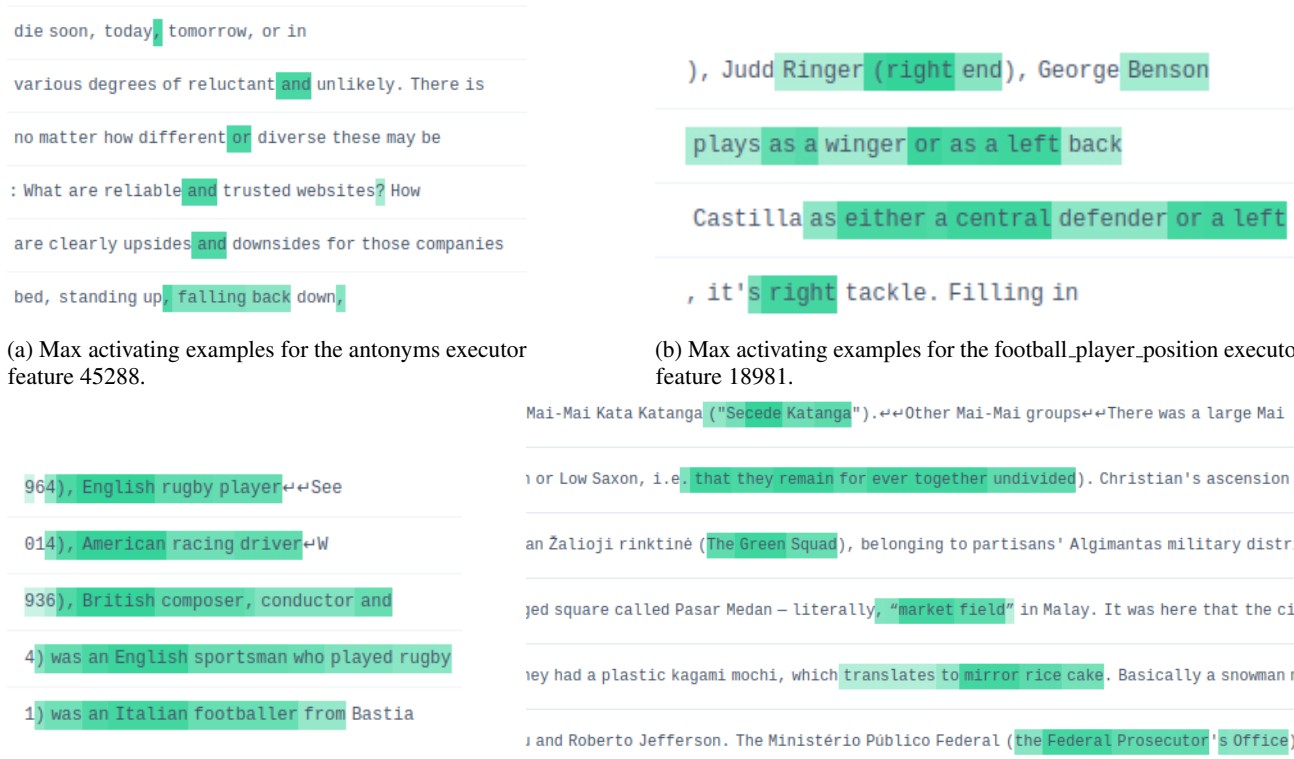

(a) Max activating examples for the antonyms executor feature 45288.

(b) Max activating examples for the football_player_position executor feature 18981.

(c) Max activating examples for the person_profession executor feature 46729.

(d) Max activating examples for translation to English executor feature 62633.

*Figure 29.* Max activating examples for Gemma 2 2B 65k executor features from Figure Figure 25b.

