# OpenReview forum: "Scaling Sparse Feature Circuits For Studying In-Context Learning"
_ICML.cc/2025/Conference — ICML 2025 poster_

### Official Review · Reviewer_1tPd · 2025-03-11

**Overall Recommendation:** 2

**Summary:**

This paper  aim to understand the mechanism of ICL by leveraging SAE, along with other techniques such as ITO and SFC, to analyze the properties of ICL task vectors. And how these techniques can be combined and improved to enhance the understanding underlying mechanisms.

**Claims And Evidence:**

I have few questions and concerns:
1. What is the main contribution of this paper?

The paper attempts to introduce a new TVC. My understanding is that this approach primarily involves continued training of a pretrained SAE specifically for ICL. If my understanding is correct, this adaptation may compromise the generalizability of SAE. If the goal is to extract features for specific tasks, why not use supervised methods instead? The authors might consider referring to benchmark techniques such as DiffMean and ReFT, as outlined in AxBench [1], which may offer more effective approaches for task feature extraction.

2. What is the motivation of using SAE to analyze task vector?

3. What empirical observations or justifications led to the decision to separate task detection and task execution vectors?

The distinction between these two components should be further elaborated, with supporting evidence demonstrating why such a separation is meaningful and necessary.

[1] AxBench: Steering LLMs? Even Simple Baselines Outperform Sparse Autoencoders.

**Essential References Not Discussed:**

N/A

**Experimental Designs Or Analyses:**

1. observations that motivate the separation of task-detection and execution vector are needed
2.  results on more models and sparsity level are needed.

**Methods And Evaluation Criteria:**

Evaluation is good to me, but results on more models and sparsity level are needed. The current experiments are primarily conducted on Gemma-1, but recent models such as LLaMA and Gemma-2 also feature pretrained SAEs with varying sparsity levels. How would different levels of sparsity in pretrained SAEs impact the conclusions drawn in this study? Additional experiments on multiple models with different sparsity settings would strengthen the validity of the findings and offer a broader perspective on the generalizability of the proposed approach.

**Other Comments Or Suggestions:**

see summary

**Other Strengths And Weaknesses:**

see summary

**Questions For Authors:**

see summary

**Relation To Broader Scientific Literature:**

This paper  investigates the task vector of in-context learning (ICL) using sparse autoencoders (SAE), and design new TVC algorithms to better assist the analysis, but the main contribution need more elaboration.

**Theoretical Claims:**

N/A (This paper do not have much theoretical analysis.)

---

> ### Author Rebuttal · Authors · 2025-04-01
>
> Hello Reviewer `1tPd`, thank you for your review. We appreciate your time and feedback, but it appears there may be some fundamental misunderstandings about our paper's focus and contributions that we'd like to clarify. **You stated: “1. What is the main contribution of this paper? The paper attempts to introduce a new TVC” but this is not the main contribution of our paper.** Instead, the main contribution of the paper is the use of SAEs to understand the circuitry behind in-context learning (ICL), as is clear from the title of our paper. This is important because our contributions are considerably more exciting for mechanistic interpretability researchers. TVC is a tool we developed to help identify which existing SAE features are most relevant to ICL tasks. Critically, TVC **does not** retrain or modify the pretrained SAE in any way - it only identifies which features in the already-trained SAE are causally relevant to ICL. Therefore, the comparison to AxBench (a paper that was released two days before the final submission deadline for ICML 2025 and therefore does not need to be addressed by us since it is contemporary work) is much less relevant to our work because our research is principally about circuits rather than steering.
>
> **On the motivation for using SAE to analyze task vectors.** Our motivation is to understand the circuit mechanisms behind in-context learning. While previous works like Todd et al. (2024) and Hendel et al. (2023) discovered task vectors, they didn't explain how these vectors are mechanistically implemented in the model. SAEs help us decompose these task vectors into interpretable features that form part of the model's computation.
>
> **On separating detection and execution features:** This separation was an empirical discovery, not an assumption, supported by multiple lines of evidence:
>
> Different activation patterns: execution features activate on arrow tokens (89.8%), detection features on output tokens (96.76%) - Tables 1 and 2
>
> Layer positioning: detection features work best in earlier layers (11), execution features in later layers (12)
>
> Causal structure: ablating detection features reduces execution feature activation (Figure 11)
>
> Task-specific steering effects: both feature types affect different tasks differently (Figures 6 and 10)
>
> Our answer to reviewer `zx2e` also contains a more thorough definition of both feature types.
>
> **On results with more models and sparsity levels:** Our Appendix includes experiments across multiple models (Gemma 1 2B, Gemma 2 2B, Gemma 2 9B, Phi-3 3B) and SAE configurations. Figures 16-19 show TVC performance across these settings, demonstrating that our approach generalizes well. These figures include different sparsities, widths and models, including Gemma Scope SAEs.

---

### Official Review · Reviewer_fzas · 2025-03-11

**Overall Recommendation:** 2

**Summary:**

This paper uses sparse autoencoders (SAEs) to study in-context learning (ICL), specifically focusing on ICL tasks that can be abstracted into a task vector. The authors propose task vector cleaning (TVC), a methodology to identify task-execution features from the set of SAE features that implement the ICL tasks during generation. The authors adapt the sparse feature circuit (SFC) methodology starting from these features, demonstrating evidence of the faithfulness and task specificity of discovered circuits. This also reveals another set of features, termed as task-detection features, which are activated prior to the task-execution ones and have a downstream activation effect on them. Experiments are mainly launched on Gemma-1 2B.

**Claims And Evidence:**

Generally yes, however the contributions and claims should be clarified, and the experimental results should be approached with caution, as follows.
- The authors claim in the first contribution that the studied **Gemma-1 2B** has “10 – 35x more parameters than prior models in comparable, circuits-style mechanistic interpretability research”. However, [1] has already performed circuit-style analysis on multiple-choice question-answering tasks using models up to **70B** parameters. Thus, the claim is invalid.
- The paper lacks a detailed, quantitative description of the experimental setup throughout the manuscript, and main results are constrained to Gemma-1 2B. This affects the scope of the paper on whether the claim holds true for more general ICL tasks and larger models.

[1] Does Circuit Analysis Interpretability Scale? Evidence from Multiple Choice Capabilities in Chinchilla

**Essential References Not Discussed:**

I have not found essential missing references, however I have one concern as in Claims and Evidence, point 1. This reference was cited but not discussed in the manner described.

**Experimental Designs Or Analyses:**

Yes. I have several concerns as follows.
- **The scope and applicability of the proposed method.** The experiments presented in the main text used Gemma-1 2B SAE trained by the authors. Given the current open-sourced SAEs on larger models (Gemma Scope, Llama Scope etc) and the claimed simplicity (“For Gemma 1, it stops at 100-200 iterations, which is close to 40 seconds at 5 iterations per second.”), I assume it is plausible for the authors to launch the same series of experiments on larger scale LLMs to demonstrate the scalability.
- **Lack of quantitative description of experiments.** For example, in Figure 2, the metric “Average relative loss change” is not described in the main text until Appendix D.1. This issue also applies to the same quantity in Figure 3, and the quantities displayed in heatmaps (Figure 6, 10, 11) among others. Numerical results should be described, or at least captioned, precisely to accurately reflect the objective of interest, facilitating the interpretation and understanding of the outcome. This statement also applies to SFC-related experiments. Could you explain the normalizing and clipping process when producing the heatmap in more detail? Further, when calculating the token type activation masses to produce Table 1, what dataset are you using, and how large is the batch?
- **Missing results though reported.** This connects to point 1. The authors mentioned “...we successfully applied … to Gemma 2 2B and 9B models… it was also successful with the Phi-3 3B model.” However, I can only find the Gemma 2 2B results in the appendix, and the results with all other models are **partial** (only the TVC comparison to inference-time optimization, ITO). How do the steering results appear for these models? Will the pattern shown in the heatmaps persist?
- **Mismatched results between heatmap and text description.** In Section 3.1, task-execution features are extracted via the proposed TVC to filter out the originally noisy features. However, the features displayed in heatmaps, such as in Figure 6, are **not** the task-execution set. The feature that has the most significant effect on en_fr and en_it is eliminated through the TVC process according to Figure 22. Thus, the description of the heatmap is improper - these non-task-execution features should be highlighted in the main text instead of in the appendix. Do you have an explanation for why certain important features are discarded by TVC? Furthermore, how does this impact the trustworthiness of TVC for task-execution feature extraction?

For other concerns, see Questions For Authors.

**Methods And Evaluation Criteria:**

Yes. The task vector is a widely accepted methodology to understand and steer ICL behavior in LLMs, and the proposed decomposition of task vectors by SAE features and the subsequent study using SFC are well-inspired by current literature. However, there are concerns regarding writing and ablation studies, which affect the scope of evaluation. Please refer to **experimental designs or analyses.**

**Other Comments Or Suggestions:**

- **Please use the correct template with line numbers clearly shown on the sidebar.**
- Section 2.1: "...with f denoting the pre-activation features..." (This should be post-activation since the activation \sigma is included.)
- Section 3.1: "Figure Figure 2 presents..." (Duplicate text)

**Other Strengths And Weaknesses:**

The writing of the paper can be improved. In addition to the issue of imprecise descriptions in the evaluation, there are other points that can be addressed. For example, some notations (e.g., d_SAE in Section 3.1) and important reference methodologies (e.g., inference-time optimization, ITO, in Section 3.1) are directly used without any definition or introduction. Further, there are also mixing usage of \citet and \citep throughout the paper.

**Questions For Authors:**

My questions are as follows.
- In Section 3.1, you found that task vectors are out-of-distribution to the SAE. Since a task vector is effectively an average over in-distribution vectors, have you tried to establish a baseline by feeding these iid inputs to the SAE, and check if there are commonly-activated features? Can the task-execution and task-detection features be identified through this method? If not, what is the ratio that these two sets of features are activated respectively, when these original inputs are fed into the model?
- As described, when launching the proposed TVC, you stated it “begins with collecting residuals for task vectors using a batch of 16 and 16-shot prompts”, and then optimize the coefficients “on a training batch of 24 pairs, with evaluation conducted on an additional 24 pairs.” Have you tested how robust TVC is to these hyperparameters? Can TVC recover the task-execution features with fewer prompts when constructing the task vector? How does the quality of identified features vary when the size of the training batch changes?
- In Section 4.1.3, you mentioned, “we opt for zero ablation since it better aligns with the sparse nature of SAE features.” How do different ablation choices here affect your experimental results? Will it lead to a detrimental effect?
- The result in Section 4.1.3 appears contradictory to the claims made in the main text. For example, the circuit in Figure 12 only consists of **5 features**, while SFC requires **~500 features** to achieve a faithfulness level of 0.6. Do you have an explanation for this phenomenon? What is the functionality of the other features, given that the discovered task-related features only occupy a tiny fraction of this circuit?

**Relation To Broader Scientific Literature:**

This paper studies ICL via SAE, proposes to decompose a task vector through SAE features to locate task-related crucial features that allow for steering.

**Theoretical Claims:**

There are no theoretical claims in this paper.

---

> ### Author Rebuttal · Authors · 2025-04-01
>
> Hello Reviewer `fzas`. Thank you for your impressively extensive and thoughtful feedback, and for your careful attention to details even in the Appendix.
>
> **On our "10-35x more parameters" claim:** You note that "[1] has already performed circuit-style analysis [...] using models up to 70B parameters." We should have been more precise in our claim. By "comparable, circuits-style mechanistic interpretability research," we specifically meant work that provides end-to-end descriptions of task performance using SAEs. The Lieberum et al. paper focuses on attention heads' impact on final logits rather than providing a comprehensive explanation of MMLU processing. They state: "Our main focus in this work was on the final parts ... the rest of the circuit is still undiscovered." We will edit this into our manuscript.
>
> Experimental design concerns:
>
> **On "lack of quantitative description of experiments"**: We moved many implementation details to the Appendix due to space constraints. We'll improve figure captions in our revision to better explain used metrics like the Average relative loss change you mentioned. The normalization process for the heatmaps (Fig. 6, 10, 11) follows these steps (Appendix F):
> 1. Calculate raw metric: metric[task, feature] = steered_loss[task, feature] / loss[task]
> 2. Clip: metric[task, feature] = clip(metric[task, feature], 1)
> 3. Normalize per task:
>    metric[task, feature] = (metric[task, feature] - min_f(metric[task, f])) /
>                           (max_f(metric[task, f]) - min_f(metric[task, f]))
> 4. Threshold: metric[task, feature] = 0 if metric[task, feature] < 0.2
>
> **On token mass calculation:** We use the same parameters as for TVC: a batch of 32 prompts each with 20 randomly sampled ICL examples.
>
> **On "missing results though reported":** You cite that we mentioned "...we successfully applied ... to Gemma 2 2B and 9B models... it was also successful with the Phi-3 3B model." This statement refers specifically to the TVC algorithm's ability to reduce feature count while preserving task vector effects, not to the full suite of steering experiments. Our Appendix contains TVC results for all mentioned models (Figures 17-18), confirming this capability. Full steering experiments require more computational and human resources than just TVC. We decided to limit this type of analysis to Gemma 2B models, but expect that it will extrapolate to other models, judging by some of the max activating examples we analyzed.
>
> **On "mismatched results between heatmap and text description":** You noted that "the feature that has the most significant effect on en_fr and en_it is eliminated through the TVC process according to Figure 22."
>
> Feature 26987 is a generic "English to foreign language" translation feature. Since Spanish tokens are prevalent in model training data, the model handles English-to-Spanish translation well, while English-to-French/Italian ICL performance is weaker. Task vectors without cleaning perform poorly for these language pairs.
>
> Task vectors contain multiple mixed-language features (26987, 26594, 6594), with 26987 having the cleanest translation examples. Since en_fr and en_it directions aren't naturally strong in the model, TVC reconstructs them using a different mix of features.
> In better-trained models and SAEs (Figure 25), these tasks have separate execution features. This shows that TVC finds a decent performing sparse combination of features in the absence of strong task-specific features.
>
> Questions:
>
> **1. On task vectors being ood:** Yes, we tried identifying common features across individual residual streams, but this highlighted many dense features alongside task-relevant ones. For Gemma-1 2B SAEs, individual residual streams activate ~40 features versus ~10 in averaged task vectors and 3-5 after cleaning.
>
> **2. On robustness to sample sizes:** We briefly explored different batch sizes and shot counts. Larger samples didn't significantly improve the TVC results. Smaller shot counts risk missing task-specific features whose presence is weaker early in the prompt. Smaller batch sizes may lead to extracting batch-specific features, especially for tasks with weaker vectors. Strong task-executing features are extracted even from small batches.
>
> **3. On ablation:** Our experiments showed no dramatic differences between zero and mean ablation. Zero ablation also doesn’t require us to store SAE activation statistics.
>
> **4. On the apparent contradiction in Section 4.1.3:** Figure 12 shows only a small subset of the circuit focusing on core task-related features. The ~500 features needed for 0.6 faithfulness include generic pattern repetition features, task executing features split across multiple features and layers, and features encoding answers in later model layers.
>
> Overall, we appreciate your feedback on the specificity of our claims, and will clearly state the takeaways from our research in our manuscript. We hope you will revise your score in light of this!

---

> > ### Comment · Reviewer_fzas · 2025-04-04
> >
> > Thanks for your response to my questions.
> >
> > **Scope and applicability:** This part is not addressed. To me, TVC plots in Figures 17-18 are insufficient since these do not necessarily connect to the steering results. My concern is that providing results on a **single** model is insufficient to draw a conclusion on (vector-abstracted) general in-context learning.
> >
> > **Mismatched results:** My concern here is two-fold. First, TVC discards this (multi-lingual) strongest translation feature candidate according to the heatmap. This brings questions on the **validity and effectiveness** of TVC: why and how could it ignore the strongest candidate? If the interpretation is that TVC tends to reconstruct the task vector by **ignoring strong, broader usage features**, then this is a critical limitation that should be mentioned and emphasized. Second is that the current heatmap is **misleading**: the caption does not match the demonstrated result, in the sense that (1) several features with high impact are not included in the task-execution set, and (2) several features with low impact are included in the task-execution set. This phenomenon should be depicted in the plot and be properly discussed.
> >
> > **Task vector and SAE feature identification:** As in my raised concern, can you further provide the **ratio** that task-execution and task-detection features are activated, when you feed your ICL inputs to the model? This is a baseline to check if these two sets of features are indeed critical for the ICL task such that they are constantly frequently activated; if not, the validity of these sets should be questioned.
> >
> > **Contradiction in Section 4.1.3:** I understand that in Figure 12 only a tiny fraction is illustrated. However, given this significant gap between the claimed number of effective task-related features and the number of total features, the ablation on whether there are features hidden in the circuit that can also steer model performance is necessary. If all other features are generic but not ICL-specific, it would strengthen the conclusion on TVC; otherwise, it would weaken the result.

---

> > > ### Author Response · Authors · 2025-04-09
> > >
> > > **Scope and applicability**
> > >
> > > We agree that our TVC results on other models do not show that our findings necessarily generalize to larger models. However we disagree that this is an issue we need to address in this paper: it is already extremely packed with details and adding more work that would likely need to be in an Appendix would make the paper more confusing for other readers. While we understand this concern, we note that seminal works in circuits mechanistic interpretability such as the IOI Circuit paper [1] have also focused on single, smaller models than Gemma 2B and have made significant contributions to the field. We expect our work to inspire future work to study the generalization of our findings to larger models.
> > >
> > > [1] Interpretability in the Wild: a Circuit for Indirect Object Identification in GPT-2 Small
> > >
> > > **Mismatched results**
> > >
> > > **Regarding translation tasks:** As you noted, this is a task-execution feature candidate for translation tasks. Since it has a strong effect across all translation tasks, it is by no means task-specific. Steering with just this feature produces a much lesser effect than the fully cleaned task vector of 3 features.
> > >
> > > It's also worth noting that Spanish is much more broadly represented in the training data, and Gemma 1 2B demonstrates stronger capabilities for English-to-Spanish translation compared to other languages. Therefore, it's not surprising that a better English-to-French direction can be built without this feature.
> > >
> > > We do not believe that this is a critical limitation of the method, since its main purpose is to discover task-specific features in the task vector, if there are any present. We would only expect it to discover features like 26987 in a generalized translation task vector.
> > >
> > > **Regarding the heatmap caption:** It's important to note that directly comparing effect magnitudes across different tasks on the normalized heatmap is not straightforward. Different tasks have different starting losses, different task vector effectiveness, and different model capabilities. For example, a task that's well-represented in the model's knowledge will have lower initial loss and thus a smaller potential for improvement from any feature.
> > >
> > > This is precisely why we normalize effects task-by-task. The normalization allows us to identify which features have the strongest relative effect for each specific task rather than making cross-task comparisons of absolute effect sizes.
> > > Our captions state that "most tasks have a single feature with a high effect on them, and this feature generally does not significantly affect unrelated tasks" and "Most features boost exactly one task, with a few exceptions for similar tasks like translating" Although most of the tasks have a feature that has a noticeably stronger effect on it than others, they often have weaker but still task-specific ones too. We will change it to “most features that have a strong effect are highly task specific”.
> > >
> > > **Task vector and SAE feature identification**
> > > We ran activation fraction experiments (batch 32, 24 n-shots) for detectors and executors.  For each task, we measured the percentage of ICL pairs where the top features from heatmaps were active (on corresponding tokens), split into "head" (first 4 examples) and "tail" (remaining 20) activations. We measured in-task and cross-task percentages.
> > >
> > > Results for executor features:
> > > - In-task activation: 50% (head), 79% (tail)
> > > - Cross-task activation: 14% (head), 17% (tail)
> > >
> > > Results for detector features:
> > > - In-task activation: 42% (head), 40% (tail)
> > > - Cross-task activation: 3% (head), 3% (tail)
> > >
> > > These results strongly validate our claims: both feature types show significant task-specificity (much higher in-task than cross-task activation). The increasing activation rate for executors from head to tail demonstrates they become more active as the model observes more examples of the task. Lower percentages for detector features could be explained by them being more prompt-specific than executors on average.
> > >
> > > **Contradiction in Section 4.1.3**
> > > When restricting IE-based node ablation to just layers 11-12 (where we study executors and detectors), we need far fewer features—approximately 10 on average to achieve 0.6 faithfulness. Interestingly, in these cases, faithfulness often exceeds 1 because removing certain negative-IE features actually improves model performance beyond the baseline, so we had to clip faithfulness to 1. If we take the best metric after ablation as the non-ablated metric (to make faithfulness <= 1), we need 40-60 nodes on average.
> > >
> > > Brief manual examination revealed that among the most important features are attention output and residual stream features like those shown in Figure 12, as well as several transcoder features with activation patterns similar to executor features. A strong “->” token feature was also among them.
> > >
> > > We hope these clarifications address your concerns and demonstrate the validity of our approach and findings.

---

### Official Review · Reviewer_zx2e · 2025-03-14

**Overall Recommendation:** 2

**Summary:**

The paper applies sparse autoencoders (SAE) to better understand in-context learning (ICL).

The paper first learns SAE representations of task vectors. Task vectors are first constructed in a heuristic manner (averaging the residual streams of arrow tokens (between inputs and outputs). The paper proposes "task vector cleaning," in which the SAE decomposition is fine-tuned on the ICL task with a sparsity constraint. This improves the zero-shot task performance as well as sparsity. Here, "task execution features" are discovered, which activate directly before an output. Steering experiments are conducted, which establish the causal effect of task latents on the task. Similar tasks have similar strong steering vectors.

Then, a circuit learning approach is applied. The primary finding in this section is the identification of "task detection features," which activate on output tokens in the training data. It is shown that detection features causally activate executor features.

**Claims And Evidence:**

The primary claim is that sparse autoencoders can help understand in-context learning circuits. This is supported by the identification of single SAE features that causally effect performance in the in-context learning task (Figure 6). The claim that this helps with understanding is backed up by the identification of "detection" and "execution" features. The evidence here is somewhat less convincing, primarily because the definitions of these features are not clearly stated. For example, while Table 1 shows that executor features activate mostly on arrow tokens, it is unclear if this is remarkable given that executor features are "characterized by" the property that "Their activation peaks on the token immediately preceding task completion." A more comprehensive analysis of what these features are for each task would be beneficial.

**Essential References Not Discussed:**

NA

**Experimental Designs Or Analyses:**

Experiments generally seemed valid. However, as mentioned above, definitions of detection and execution features were underspecified.

**Methods And Evaluation Criteria:**

The methods seem solid. The task vector cleaning method in particular is a simple approach that appears to work well to get strong SAE decompositions.

**Other Comments Or Suggestions:**

NA

**Other Strengths And Weaknesses:**

Overall, the paper pursues a promising direction in applying sparse autoencoders to better understand in-context learning. The development of task vector cleaning appears to be a clean way to learn task vectors that are cleanly expressed by SAE latents.

The primary weakness of the paper is a lack of specificity when describing and analyzing the proposed detection and execution features. Without a more precise understanding of how these features are characterized, it is difficult to assess the validity of this abstraction. Here, the most convincing evidence of this abstraction would be a demonstration of (1) the ability to construct task vectors on new tasks based on learned features, similar to how it has been demonstrated that LLM behavior can be manipulated through SAE latents, and (2) a more precise connection between detection and execution features and behavior that distinguishes their specific roles.

**Questions For Authors:**

NA

**Relation To Broader Scientific Literature:**

The paper contributes to a growing literature in mechanistic interpretability, in particular in using sparse autoencoders. While sparse autoencoders have been shown to identify interpretable and causal features in LLMs, there use to study more specific behaviors and capabilities is very nascent. This paper considers the task of in-context learning, which is a topic of significant interest. The existence of detection and execution features are an interesting finding.

**Theoretical Claims:**

NA

---

> ### Author Rebuttal · Authors · 2025-04-01
>
> Hello reviewer `zx2e` – thank you for your thoughtful review. We appreciate your recognition that our methods are "solid" and that task vector cleaning is "a simple approach that appears to work well to get strong SAE decompositions."
>
> **Regarding feature definitions**: we consider *task-specific causal influence to be the defining characteristic of both detector and executor features*, and apologize for not making this clearer in the main text. We will update the manuscript based on this feedback. To expand specifically on what we mean by task-detection and task-execution features:
>
> **Task-detection** features are defined as *intermediate circuit components between the raw prompt inputs and the executor features*. They function as a specialized probing mechanism that identifies and encodes which specific task is being performed. These features primarily activate on output tokens (96.76% as shown in Table 2), consistent with their role in monitoring completed examples to determine the pattern being demonstrated. Importantly, our steering experiments in Section 4.2 demonstrate that these detector features also have direct causal effects on task performance when steered on blank output tokens.
>
> **Task-execution** features are defined as features that *directly impact task completion*. They affect whether the specific task happens without requiring additional intermediate processing. Their strong activation on arrow tokens (89.8% as shown in Table 1) demonstrates their positioning at precisely the point where the model must apply the identified transformation.
>
> For greater specificity, we classify these features using clear criteria:
> - **Detection features**:
> (1) consistent activation on output tokens in examples, (2) minimal activation during generation, (3) causal influence on executor features (quantified in Figure 11), and (4) direct causal effect on task performance when steered on output tokens
> - **Execution features**: (1) peak activation on arrow tokens immediately preceding task completion, (2) strong causal effect on task performance when steered with (Figure 6), and (3) task-specific activation patterns on raw data.
>
> **Regarding your suggestion about "constructing task vectors on new tasks"** - if you're referring to zero-shot generalization, our steering experiments in Section 3.2 and Figure 6 demonstrate the causal effect of our identified features on model behavior for the tasks studied. While running additional experiments on entirely new tasks is beyond our current rebuttal time frame, we believe the consistent patterns across our current task set suggest potential for generalization.
>
> The circuit analysis in Section 4.2 and Figure 11 provides empirical validation of our proposed abstraction by showing that: (1) ablating detection features reduces execution feature activation, (2) this relationship is consistent across tasks. These quantitative measures establish that detection and execution features form distinct functional components in the ICL circuit.
>
> We appreciate your feedback and will incorporate these clarifications to strengthen the paper.

---

### Official Review · Reviewer_K3Cy · 2025-03-15

**Overall Recommendation:** 3

**Summary:**

This paper explores how sparse autoencoders (SAEs) can enhance our understanding of in-context learning (ICL) mechanisms in LLMs. The paper's main contributions include:
- Identifying two core components of ICL circuits: task-detection features that identify required tasks from the prompt, and task-execution features that implement those tasks during generation.
- Developing a Task Vector Cleaning (TVC) algorithm that decomposes task vectors into their most relevant sparse features, enabling more precise analysis of ICL mechanisms.
- Adapting the Sparse Feature Circuits (SFC) methodology to work with the much larger Gemma-1 2B model (30x larger than models in previous circuit analysis studies) and applying it to the complex task of ICL.
- Uncovering the interaction between these components: attention heads and MLPs process information from task-detection features to activate appropriate task-execution features.
- The authors provide evidence that these task vectors can be represented as sparse sums of SAE latents, and they demonstrate causal relationships between detection and execution features through circuit analysis.

**Claims And Evidence:**

The claim that task vectors can be decomposed into sparse, interpretable features is supported by the TVC algorithm results, showing the algorithm can reduce the number of active SAE features by 70% while maintaining or improving effect on loss.
The identification of task-detection and task-execution features is supported through thorough analysis of activation patterns, steering experiments, and ablation studies that demonstrate their causal roles.

**Essential References Not Discussed:**

The paper could have benefited from more discussion of how their findings relate to theoretical models of IC

**Experimental Designs Or Analyses:**

The experimental designs are sound:
- The token position categorization approach for handling ICL prompts is well-motivated and effective for isolating different functional roles.
- The steering experiments with both positive and negative steering provide complementary evidence for feature functions.
- The ablation studies establish causal relationships between detected circuit components.

One minor issue is that the authors restricted their circuit search to intermediate layers 10-17 of the 18 total layers, justifying this based on IE approximation quality in earlier layers. While reasonable, this limitation means some potential early-layer mechanisms might have been missed.

**Methods And Evaluation Criteria:**

The methods and evaluation criteria used in this paper are appropriate for the research questions:
- The Task Vector Cleaning algorithm effectively isolates task-relevant features
- The steering experiments provide measure of causal influence by testing how individual features affect performance on specific tasks.

**Other Comments Or Suggestions:**

It would be interesting to see how the detected ICL circuits relate to other mechanisms that have been studied in language models, such as factual recall or logical reasoning.

**Other Strengths And Weaknesses:**

Strengths:
- The paper successfully scales mechanistic interpretability techniques for ICL.
- The identification of two distinct mechanisms (detection and execution) provides a clear conceptual framework for understanding ICL.
- The paper demonstrates how SAEs can be used for more than just interpreting individual features but for understanding complex model behaviors.
- The TVC algorithm is a valuable contribution that could be applied to other research on task vectors.

Weaknesses:
- The analysis focuses on relatively simple ICL tasks (e.g., antonyms, translation). It's unclear how well the approach would extend to more complex reasoning tasks.
- The authors note that there are often multiple competing execution and detection features, suggesting redundancy in the model's representations that could complicate interpretation.
- The paper doesn't explore how these mechanisms might differ across model

**Questions For Authors:**

You mentioned person_profession and present_simple_gerund showed unusually weak detection-execution connections. Do you have hypotheses about why these specific tasks might be processed differently?

**Relation To Broader Scientific Literature:**

The paper effectively situates its contributions within SAEs and ICL literature:
- It builds upon prior work on task vectors (Todd et al., 2024; Hendel et al., 2023) by providing a mechanistic explanation of how these vectors function through sparse features.
- It extends the SFC methodology from Marks et al. (2024) to handle more complex tasks and larger models.
- It connects to work on induction heads (Olsson et al., 2022) while demonstrating that ICL requires more complex mechanisms beyond just induction.

**Theoretical Claims:**

There is no proofs or theoretical claims

---

> ### Author Rebuttal · Authors · 2025-04-01
>
> Hello reviewer K3Cy,
>
> Thank you for your thoughtful and supportive review of our paper. We appreciate your detailed feedback and are grateful for your recommendation to accept the paper.
>
> Regarding your specific question about the weak detection-execution connections for the `person_profession` and `present_simple_gerund` tasks:
>
> For `person_profession`, our analysis revealed the model's general performance of this task is relatively poor. As seen in Figure 27f, the strongest detector feature is a single token feature focused on journalism, indicating that SAE features did not pick up strong task-related directions alongside poor model's accuracy. This both contributes to weak connection of executing and detecting features.
>
> For `present_simple_gerund`, the situation is more nuanced. One possible explanation is feature splitting. Since gerund forms are common in the training data, they have several corresponding detector and executor features. At the same time, executor and detector features with the strongest steering effect may correspond to different cases of gerund occurrences, and thus be weakly connected.
>
> Thank you again for your positive feedback on our work!

---

### Decision · Program_Chairs · 2025-05-01

**Decision:**

Accept (poster)

**Comment:**

This paper feels truly borderline. While the reviewers find merit in many of the ideas, they also identify some weaknesses that should be addressed.

As I have read the paper and considered the reviews, I feel that the strengths outweigh the weaknesses. I think this paper is thoughtful, well-written and contributes interesting insights; I believe that it represents a positive contribution to the field. The TVC algorithm is particularly interesting.

I strongly encourage the authors to incorporate the reviewer feedback. In particular, I caution the authors against overclaiming, given the limited nature of the tasks and models considered.